# Microseismic Signal Characteristics of the Coal Failure Process under Weak-Energy and Low-Frequency Disturbance

Xiaoyuan Sun [1,2], Yongliang He [1,2,*], Tingxu Jin [1], Jianlin Xie [1], Chuantian Li [1,2] and Jiewen Pang [1,2]

[1] College of Safety and Emergency Management Engineering, Taiyuan University of Science and Technology, Taiyuan 030024, China; sunxy253@126.com (X.S.); jintingxu1997@163.com (T.J.); jianlinx@126.com (J.X.); 2009013@tyust.edu.cn (C.L.); pjwfanfan@126.com (J.P.)

[2] Intelligent Monitoring and Control of Coal Mine Dust Key Laboratory of Shanxi Province, Taiyuan University of Science and Technology, Taiyuan 030024, China

\* Correspondence: hylust@163.com

**Abstract:** In deep mining, "critical static stress + slight disturbance" is an important inducing form of coal mine rockburst disasters. In previous studies, the critical static stress has been shown to be consistent with the loading direction of a slight disturbance but cannot reflect all types of rockbursts. In addition, the research that uses microseismic (MS) signals to reflect the overall process and critical stages of coal failure and instability under weak-energy and low-frequency disturbance conditions is immature, and more information, such as the critical state, has not been fully revealed. The aims of this paper are to further elucidate the important role of weak-energy and low-frequency disturbances in the occurrence of rockburst disasters. First, briquette samples were prepared from the Tashan Coal Mine, which is severely affected by rockbursts, and their homogeneity was verified using ultrasonic longitudinal wave velocity. Second, the natural frequency of the coal sample specimens was measured using a testing system. Then, based on the self-developed static pressure loading system, dynamic and static combined loading test system and MS signal monitoring device, the MS signal characteristics during the process of coal body failure and instability were comprehensively analysed. Finally, a comparison was made between weak-energy and low-frequency disturbances and impact disturbances. The results are summarized as follows. (1) The longitudinal wave velocity test results reflect that the briquette samples prepared in the experiment have high homogeneity. The smaller the particle size is, the higher the density and moulding pressure, and the denser the sample. (2) The natural frequency of the briquette samples is between 30.79 Hz and 43.34 Hz, and most of them fluctuate at approximately 35 Hz. (3) During the static loading stage, the occurrence of more than three MS signals of larger magnitude in a continuous cluster is an important criterion for the critical failure of the samples. (4) The weak-energy and low-frequency disturbance actually leads to fatigue damage, and the briquette sample experiences three stages: the near-threshold stage, the high-speed expansion stage and the final fracture stage. The smaller the particle size of the coal sample, the denser the specimen, the stronger the amplitude and energy of the single effective MS signal formed during the destruction process, the longer the time duration of crack expansion from the near-threshold stage to the high-speed expansion stage, and the stronger the ability of the coal sample to resist weak-energy and low-frequency disturbances. This study may contribute to a more comprehensive understanding of the destabilization mechanism of coal bodies and MS signal characteristics under weak-energy and low-frequency disturbances and provide a reference for further research and discussion.

**Keywords:** rockburst; weak-energy and low-frequency disturbance; dynamic and static combined loading; coal instability; microseismic signal

## 1. Introduction

As a typical dynamic disaster, rockburst poses a great threat to the safety of personnel and equipment in all mining countries. Bennett and Marshall [1] and Wu et al. [2] mapped

the distribution of typical mines in the world where rockbursts have occurred. In the United States, 312 rockbursts were counted from 1936 to 2013 [3,4]. The Czech Republic and Poland had 122 deaths due to rockbursts from 1983 to 2003 [5,6]. Rockburst is an important cause of death for coal mine operators in South Africa [7]. In addition, cases of rockburst have been reported in Canada [8], Australia [9], Germany [10], Russia [11] and other countries. China is also one of the countries that is most seriously affected by rockburst disasters worldwide [12]. In recent years, with the increasing demand for resources, an increasing number of mines have gradually deepened. This has led to an increase in initial stress and significant changes in the mechanical properties of coal and rock, as well as increasingly complex hydrogeological and mining conditions, resulting in increasingly severe accidents [13,14]. Furthermore, rockbursts may produce gas-dynamics phenomena, instantly changing the gas parameters of the mine [15], damaging the ventilation equipment [16] and even triggering secondary accidents such as mine fires [17–19], floods [20] and gas disasters [21,22]. Therefore, it is urgent to conduct research on the effective prevention and control of rockbursts in deep mines.

In the study of rockburst, clarifying the mechanism of its occurrence is a crucial prerequisite. However, due to the complexity, suddenness and diversity of disasters, no universally applicable conclusion has been formed thus far. Generally, it is believed that during deep mining, coal and rock will face the complex mechanical environment of "three highs and one disturbance", i.e., the combined effects of high geopathic stress, high geothermal temperature, high karst water pressure and mining disturbance [23]. For rockburst, high ground stress and mining disturbance play an important role [24]. On this basis, Li et al. [25] summarized the triggering modes of coal-rock destabilization and rockburst into two dynamic–static combination loading modes, i.e., "elastic static stress + impact disturbance" and "critical static stress + slight disturbance", and provided their respective laboratory characterization methods. In coal mines, the relative energy of operations such as rapid excavation and blasting is large, which can cause rapid changes in stress distribution and instantaneous instability of the coal-rock system, so it is defined as impact disturbance or strong disturbance. Due to the high frequency and intensity of rockbursts induced by strong disturbances, this topic has gradually become a hot spot of mechanism research [26–28]. For strong disturbances, academics mostly use the separated Hopkinson pressure bar (SHPB) for laboratory characterization and have achieved fruitful results [29,30]. To date, the SHPB has become a commonly used technique to measure the stress–strain relationship of various engineering materials under high strain rate conditions [31].

In contrast, there have been relatively few studies on "critical static stress + slight disturbance". However, reports of rockbursts induced by slight disturbances are not uncommon [32–36]. These slight disturbances, also called weak-energy disturbances, are often characterized by long duration and cyclic operation, and the few studies that have been carried out focused on applying vibration loads of different frequencies. Typical experimental methods include ultrasound [37], microwaves [38] and low-frequency mechanical vibration. Among them, low-frequency mechanical vibration has a high degree of compatibility with operations such as hand axe operation, drilling operation, support repair and large-scale equipment operation. Therefore, using low-frequency mechanical vibration to explore the induced mechanism of weak-energy and low-frequency disturbances on rockburst will be of more engineering significance. For instance, Pan et al. [39], Nie et al. [40], Li et al. [41] and Gao [42] used vibration tables of different sizes to analyse the effect of vibration on the distribution of cracks and mechanical properties of coal and rock. In fact, most traditional weak-energy and low-frequency disturbance experiments were conducted on servo testing machines. For example, using the Instron-1342 electrohydraulic servo material testing machine, Li et al. [43] designed a loading scheme of "prestatic stress + low-week fatigue disturbance" and investigated the instability and failure process of a red sandstone system under the condition of one-dimensional static and dynamic combinations. On this basis, Zuo et al. [44], Du et al. [45], Li et al. [46], Su et al. [47] and Jiang et al. [48] successively

extended the "critical static stress + small disturbance" to the two-dimensional and even three-dimensional domains. At the same time, Zuo et al. [49] designed the disturbance load as a single-period sinusoidal wave with an amplitude of 100 kN and a frequency of 2 Hz and established a catastrophe theory model for rock failure and instability [50]. In addition, Liu et al. [51] conducted fatigue loading tests on intermittently jointed rocks using an MTS 793 servo machine and systematically described the effects of cyclic loading parameters on fatigue deformation characteristics, fatigue energy and damage evolution, fatigue failure and progressive failure behaviour. Li et al. [52] used an MTS815 servo-controlled testing machine to reveal the deformation characteristics of rock under cyclic loading and established a theoretical model of abrupt change in rock damage from energy dissipation. Wang et al. [53] conducted a multistage intermittent cyclic loading test on precracked red sandstone using a WHY-CTS600 pressure testing machine and analysed the fatigue damage and fracture evolution characteristics of the rock specimens by using the acoustic emission (AE) technique and digital image correlation (DIC) method. Notably, in most of the servo machine series experiments, the dynamic and static loads of high static stress and weak-energy disturbance are applied in the same direction, i.e., along the axial direction of the cylindrical specimen. This has high compatibility with the engineering reality that the static stress and disturbance stress from the top or bottom plates combine to cause coal and rock damage and instability. However, it is difficult to reflect all types of rockbursts, especially when the direction of the critical static stress and weak-energy disturbance are inconsistent.

Taking a panoramic view of the domestic and foreign research status, the current research on impact disturbances presents characteristics of "three more and three less", i.e., more research on strong disturbances and less research on weak-energy disturbances; more research on single low-frequency vibrations and less research on the combination of static load and vibration load; and more research on high static stress and weak-energy disturbances along the same path and less research on loading in different directions. Furthermore, the mechanism of coal failure and rockburst induced by weak-energy and low-frequency disturbances is not yet clear. Obviously, the "critical static stress + slight disturbance"-induced destabilization of the coal body needs to go through two key stages. First, the coal body is already in a critical limit state under the influence of high static stress. Second, the coupling of the weak-energy and low-frequency disturbances on the basis of which the critical state is broken by several cycles of repeated actions [54]. In this process, weak-energy and low-frequency disturbances are the triggering conditions for coal body failure and instability, and whether the critical state has been formed and can continue to be maintained is the key prerequisite for the occurrence of rockburst. However, the critical state is affected by multiple factors, which are extremely difficult to determine directly. The action mechanism of weak-energy and low-frequency load parameters and physical and mechanical properties on coal-rock failure and instability is relatively complex, and it is difficult to directly relate to rockburst warning in engineering practice. Therefore, this process needs to be characterized by other means.

In engineering practice, the methods used for rockburst warning are mainly divided into traditional contact methods (such as the drilling cutting method, stress monitoring method and separation monitoring) and geophysical methods. In comparison, the latter has the advantages of being noncontact, continuity, large-area prediction and not affecting normal mining and is more widely used [55]. In fact, the failure and instability of coal and rock are prerequisites for rockburst. There are a large number of cracks and defects inside a coal and rock mass, and the deformation and failure processes are closely related to the generation, expansion, convergence, and connection of these cracks. This process inevitably dissipates various energies, such as surface energy, elastic energy, thermal energy, acoustic energy and electromagnetic energy. Meanwhile, part of the energy escapes and is effectively monitored. The destabilization process of the coal body is often accompanied by microseismic (MS) events, AE, electromagnetic radiation (EMR), surface potential transients and other acoustic and electrical signal anomalies [56]. Among these signals, the

monitoring methods for MS events are relatively mature [57], providing a theoretical basis and experimental reference for characterizing the coal failure process under weak-energy and low-frequency disturbances.

In this study, first, coal samples are prepared with a high degree of homogeneity, and an experimental platform is set up for static and vibration load coupling to clarify the mechanical response differences of a coal body under different directions of static and dynamic loading. Then, based on the MS monitoring method, the signal differences in different samples, loading methods and damage processes are analysed. Finally, the overall process and key nodes of coal failure and instability under the coupling of static stress and weak-energy and low-frequency disturbances are characterized from the perspective of signal characteristics, and the feasibility of using the above signals to reflect the process of coal and rock failure induced by weak-energy and low-frequency disturbances is explored. The research results will help to supplement and clarify the rockburst mechanism and provide scientific support for equipment development and institutional innovation.

## 2. Materials and Methods

### 2.1. Material Introduction and Specimen Preparation

The experimental specimens were taken from the 8204 fully mechanized top-coal caving face of the Tashan Coal Mine in Datong, Shanxi Province, China. The Tashan Coal Mine is one of the largest underground coal mines in the world, with an annual production of up to 25 million tons. However, it has experienced multiple rockburst accidents [58] and been severely affected by coal-rock dynamic disasters [59]. Studies have shown that the frequent occurrence of rockbursts is closely related to the geological environment and mining disturbances in this mine [60]. We conducted a firmness coefficient (*f* value) test on the collected samples using the drop hammer method [61], and the firmness coefficient was 0. 9231, indicating that the coal in Tashan is relatively hard. To eliminate the influence of anisotropy and the random distribution of primary fissures, joints and other structures within the coal body on the test results, the specimens were finally prepared as moulded coal. Sample preparation was carried out in three steps. First, the collected bulk coal body was crushed and ground. Second, they were sieved into three different particle sizes (<0.25 mm, 0.25~0.5 mm and 0.5~1.0 mm) using a standard sieve. Third, an appropriate amount of coal tar was homogeneously mixed into the coal dust, and the blended masses of 850 g, 925 g, 1000 g, 1075 g and 1150 g were placed in the mould and compressed into an embryo of approximately $100 \times 100 \times 100$ mm using an MTS electrohydraulic servo press, set to a low-speed displacement control mode (0.2 mm/min). The compressed samples were kept under stress on the servo press for more than 12 h. As shown in Figure 1, the final prepared samples exhibited differences in powder particle size, density and moulding pressure, which were defined as TS1–15. In addition, two embryos were prepared using the same method for the static loading test, with serial numbers TS16–17.

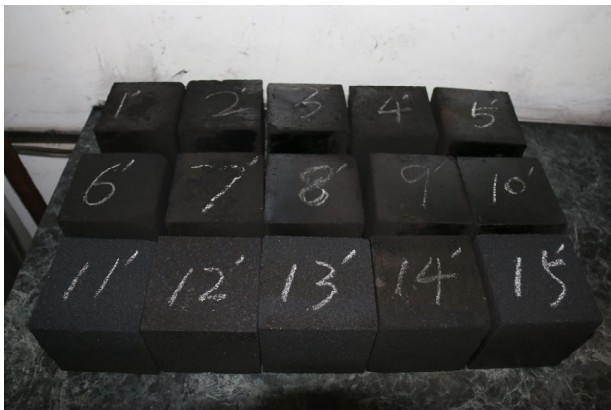

**Figure 1.** Illustration of the coal specimens.

*2.2. Test Setup and Instrumentation*

Figures 2 and 3 illustrate two important test systems required for the experiment, i.e., the natural frequency test system and the dynamic–static combination test system. The main purpose of Figure 2 is to determine the natural frequency of the coal specimen, which provides the basis for the subsequent setting of the weak-energy and low-frequency loading parameters. In this thesis, we adopt the instantaneous hammering method to determine the natural frequency of coal. As shown in Figure 2a, the natural frequency test system includes a signal excitation system and a response acquisition system. (1) The signal excitation system consists of an SD1428 force hammer, an SD1431 electromagnetic amplifier and a TST3406 dynamic test analyser. The SD1428 force hammer contains an SD1422 piezoelectric quartz mechanical transducer, which is designed to generate a shock pulse covering all frequencies from zero to infinity [62]. The SD 1431 charge amplifier is connected to the front-end of the SD1422 piezoelectric quartz mechanical sensor to provide the acceleration values of the vibration and shock excitation. The TST3406 dynamic test analyser is equipped with DAP3.14 signal acquisition software, which has a high sensitivity (trigger level of 0.1562 V) and sampling frequency (up to 40 MHz) and is mainly used for the acquisition of the excitation signals. (2) The response acquisition system includes an SF1500MEMS capacitive accelerometer, a ZDKT-1 data acquisition system and MKJC3.0 data acquisition software. The SF1500MEMS capacitive accelerometer is developed and produced by Colibrys Company, Vaud, Switzerland, and it has the advantages of a high dynamic range, wide bandwidth, a low deformation rate, high impact and thermal stability; this accelerometer is particularly suitable for the field of strong dynamic parameter monitoring. During the testing process, Vaseline is used to attach the accelerometer to the coal specimen. The ZDKT-1 data acquisition system was independently designed by China University of Mining and Technology (Beijing, China) and integrates the NI 9219 and NI 9234 data collectors developed by National Instruments (NI) of Austin, TX, USA. The former is mainly used for stress and strain measurements, while the latter provides sound and vibration signal acquisition functions such as sound measurement, frequency analysis and transient analysis. MKJC3.0 data acquisition software is a natural frequency acquisition software with real-time signal monitoring, storage and analysis processing functions.

Notably, to avoid the influence of an external medium on the dynamic parameters of the coal body and to reduce its damping ratio, as shown in Figure 2b, a soft support method with the combined effect of a polyurethane foam board and soft sponge was adopted during the natural frequency test. The force hammer is used to instantaneously strike the centre point on the side of the briquette, causing the specimen to vibrate. The response signal is attached to the capacitive accelerometer on the opposite side and transmitted to the computer through the ZDKT-1 data acquisition system. The response signal is collected by MKJC3.0 software. At the same time, the excitation signal of the force hammer is also transformed and amplified by the charge amplifier and then transmitted to the computer through the TST3406 dynamic test analyser, and the excitation signal is monitored and stored synchronously using DAP3.14 signal acquisition software.

The schematic and physical diagrams of the dynamic–static combination test system are shown in Figure 3. As seen from the figure, the test system consists of a clamping subsystem, a static load subsystem, a dynamic load subsystem and a signal recording system. The clamping subsystem has three functions: (1) to ensure the stability of the test object and to avoid the overall movement of the specimen during testing; (2) to evenly distribute the static pressure load on the coal sample; and (3) to lay a steel baffle plate between the coal sample and the vibration exciter so that the vibration wave propagates efficiently in the medium of the coal sample, and at the same time, the reflection effect occurs at the boundary [54]. The static load subsystem is composed of a separating jack and a manual hydraulic pump. The maximum pressure of the jack is 10 tons, and the maximum stroke is 100 mm, which is used to simulate the static pressure load above the coal body. The dynamic load subsystem adopts the JZ20 vibration exciter and the GF300

power amplifier developed by Yuanzheng Zhenxing Technology Co., Ltd. (Beijing, China) The JZ20 vibration exciter has a working frequency of 5~5000 Hz and a maximum output force of 200 N. The GF300 power amplifier is mated with the JZ20, with a maximum output power of 300 W, an output impedance of 0.75 Ω and a maximum distortion of 3%, which can meet the experimental requirements. In the experiment, the purpose of the dynamic load subsystem is to apply a weak-energy and low-frequency disturbance to the test object, i.e., the coal body. In mining engineering practice, the waveforms and frequencies of weak-energy and low-frequency disturbances are complex, and it is almost impossible to fully reproduce the disturbance waves on site through experimental means. However, all vibration waves can be decomposed into a series of superimposed sine wave signals with different frequencies, phases and amplitudes [63]. For this reason, the corresponding destructive characteristics of the coal body when subjected to forced vibration under the action of a certain sine wave can be analysed first, and then further exploration can be conducted based on this. The sine wave signal is supplied by an AFG3022B function signal generator produced by Tektronix in Beaverton, OR, USA. The signal generator can generate 12 different waveforms, including a sine wave covering a frequency of 1 μHz to 25 MHz, with a highly stable time reference. The signal recording subsystem consists of MS sensors and a ZDKT-1 signal acquisition system. The MS sensors are also selected from SF1500MEMS capacitive accelerometers produced by Colibrys, Vaud, Switzerland. The MS signals are stored and analysed by the ZDKT-1 data acquisition system and MKJC3.0 data acquisition software.

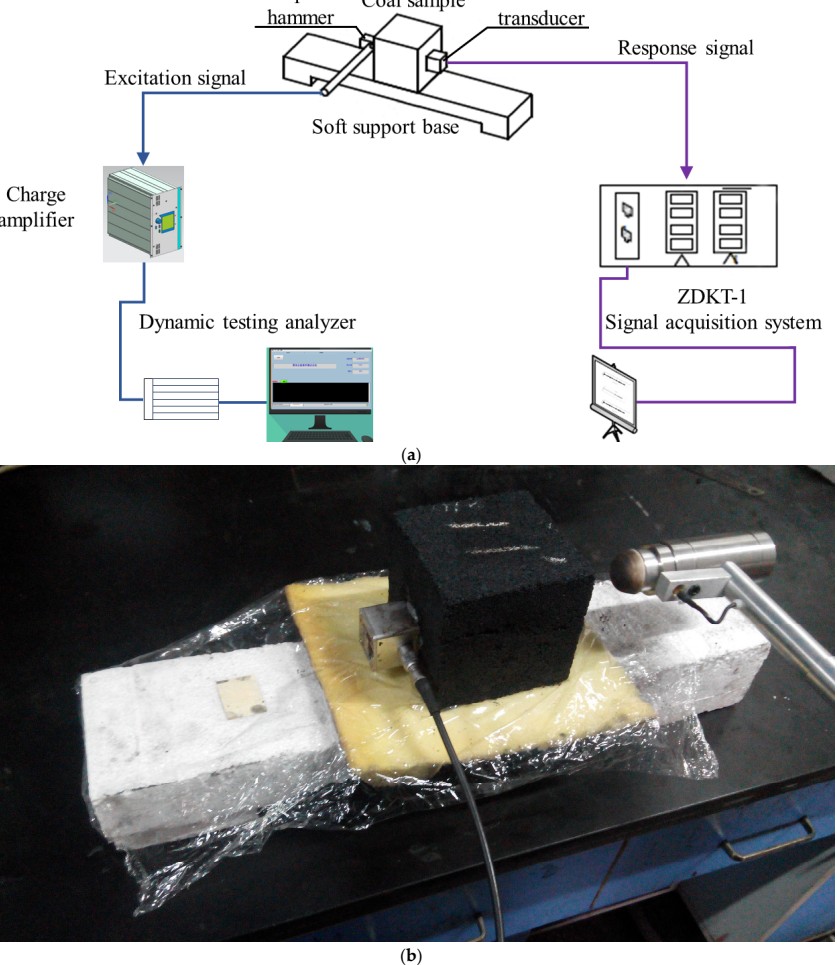

(a)

(b)

**Figure 2.** Diagram of the natural frequency testing system. (**a**) Schematic diagram; (**b**) physical diagram.

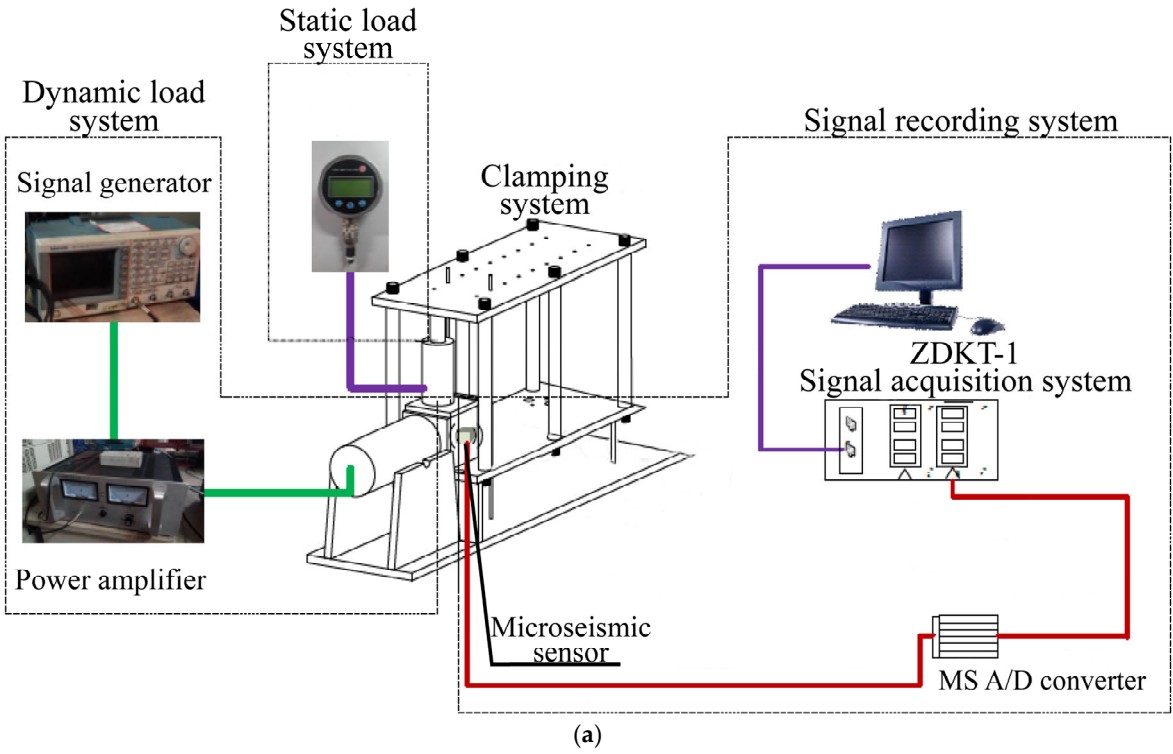

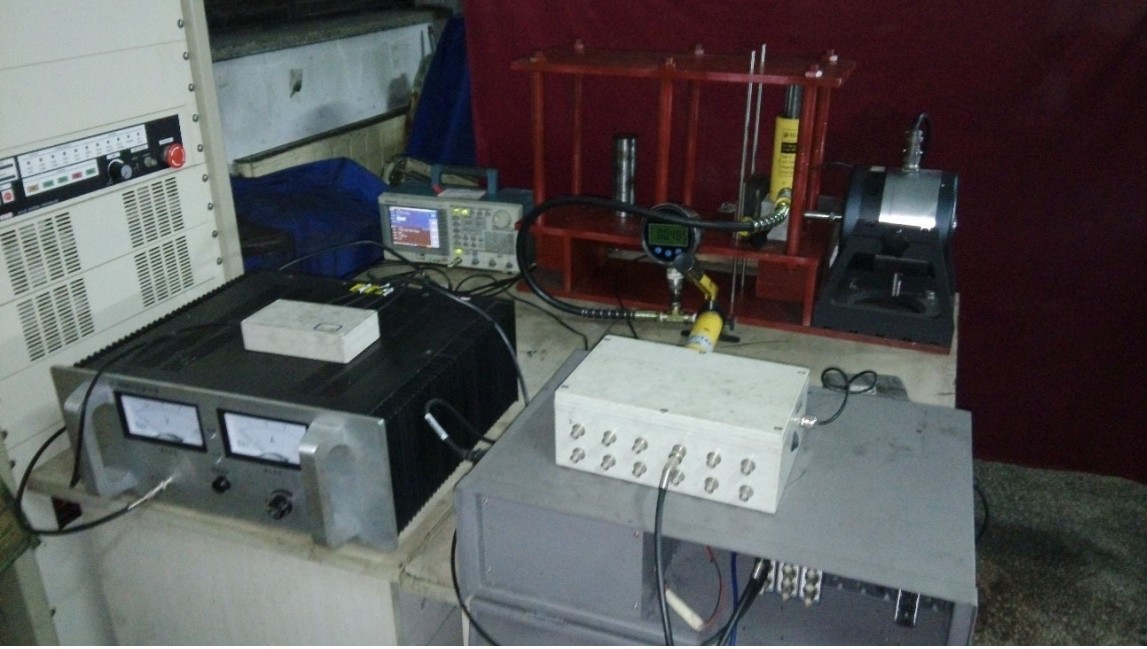

**Figure 3.** Diagram of the dynamic–static combination test system. (**a**) Schematic diagram; (**b**) physical diagram.

### 2.3. Test Condition and Procedure

The purpose of the static–dynamic load coupling experiment is to explore the influence of coal properties and disturbance parameters on the destabilization process of a specimen. Therefore, the testing is conducted in four stages, as shown in Figure 4. In the first stage, we focus on the properties of coal bodies. First, the ultrasonic longitudinal wave velocity test is performed on the coal samples to reflect the differences in parameters such as particle size, density and moulding pressure. Subsequently, the natural vibration frequency of the

briquettes is measured using the natural vibration frequency testing system (Figure 2), providing a basis for the subsequent setting of vibration parameters.

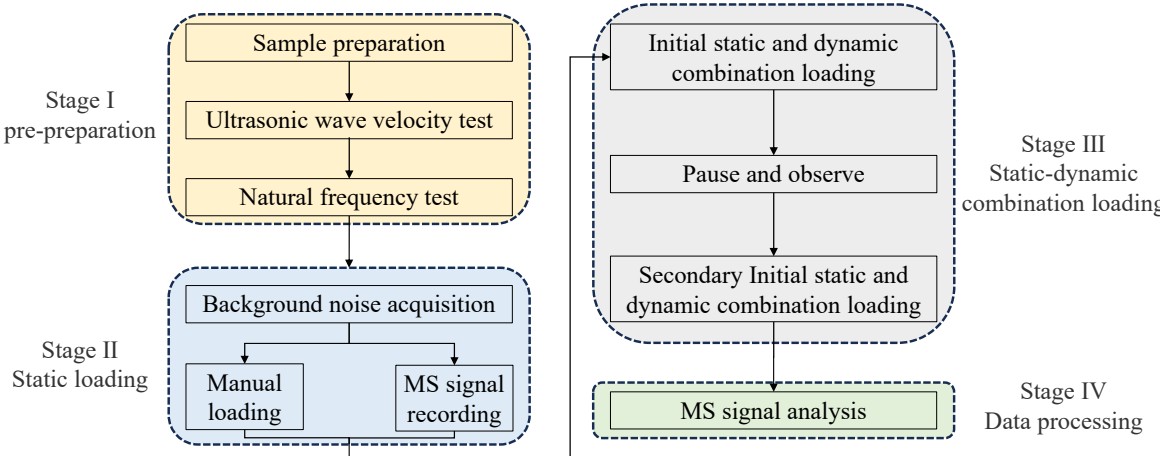

**Figure 4.** Experimental process flowchart.

In the second stage, the briquette samples TS 16 and TS 17 are selected for uniaxial compression experiments, and the MS signals in the process of coal damage and destruction are recorded. The pressure gauge readings of the static loading system and the development of the sample fissures are observed synchronously to provide the characteristics of the MS signals and the critical state criteria during the static load damage process of the coal body.

In the third stage, the excitation parameters for static and dynamic loads are appropriately adjusted, and the static and dynamic combined loading tests are conducted on the briquette samples TS1~TS15. Firstly, turn on the static load system, use the manual hydraulic pump to slowly load, and record the MS signals. When the MS signal indicates that the sample has entered the critical state, immediately stop loading, rapidly reduce the pressure of the jack by 20% and simultaneously start the loading system to perform a combination of static and dynamic loading on the sample. The frequency of the vibration exciter is set with reference to the natural frequency of the sample. The excitation is carried out in two stages, each lasting for 1 h, with a pause of 1 h during the period. The MS signals during the whole process are recorded simultaneously, and targeted processing and analysis are carried out in the fourth stage.

In the experiment, two points are highlighted: (1) In the excavation face, the coal body, which is actually disturbed by the dynamic load and destabilization phenomenon, is located near the pressure relief zone in front of the working face. This part of the coal body has experienced the process of being the original stress area–concentrated stress area–stress reduction area and has undergone a certain degree of damage. Therefore, the jack used for static loading also follows the experimental process of static loading–coal body damage–jack unloading. (2) In mining engineering practice, the Tashan Mine follows the "three-eight" work cycle, i.e., "excavation for 8 h—maintenance for 8 h—excavation for 8 h". Therefore, the vibration exciter in the experiment also adopts the loading method of "excitation for 1 h—pause for 1 h—excitation for 1 h", with the aim of closely integrating the experimental operation with engineering practice.

## 3. Test Results and Data Analysis

### 3.1. Ultrasonic Wave Velocity Test Results for the Coal Samples

To reflect the densification degree of the specimen, the ultrasonic longitudinal wave velocity test is carried out on the coal specimen first. The instrument is a ZBL-510 nonmetallic ultrasonic detector produced by Beijing ZhiBoLian Technology Co., Ltd. (Beijing, China). Its detection accuracy is ±0.05 μs, the gain amplitude is 82 dB and the reception sensitivity is less than 30 μv. Referring to the single-transmitter-single-receiver

acoustic transmission method stipulated in ASTM D 2845 [64], the transmission voltage is 1000 V, and the pulse width is 0.04 ms. The transducer is placed using the "face-to-face monitoring" method [65] using Vaseline as the coupling agent and exerting a consistent pressure on the transducer as much as possible. To eliminate the influence of external factors, each sample was tested three times in different directions, and their arithmetic mean was calculated. The final results are summarized in Table 1.

**Table 1.** Test results of the *p*-wave velocity for coal samples.

| Particle Size Grade (mm) | Sample ID | Group Number | Density (g·cm$^{-3}$) | Forming Stress (kN) | Dimension (mm) | | | Mean Value of Longitudinal Wave Velocity (km·s$^{-1}$) | | | Longitudinal Wave Velocity (km·s$^{-1}$) |
|---|---|---|---|---|---|---|---|---|---|---|---|
| | | | | | x | y | z | x | y | z | |
| <0.25 | TS1 | A1 | 0.85 | 74.45 | 102.2 | 103.5 | 102.6 | 0.402 | 0.474 | 0.449 | 0.442 |
| | TS2 | A2 | 0.925 | 101.15 | 105.5 | 102.3 | 103 | 0.482 | 0.619 | 0.612 | 0.571 |
| | TS3 | A3 | 1.0 | 189.2 | 103.8 | 106.1 | 115.4 | 0.743 | 0.736 | 0.542 | 0.674 |
| | TS4 | A4 | 1.075 | 209 | 104.6 | 105.2 | 117.4 | 0.702 | 0.670 | 0.550 | 0.641 |
| | TS5 | A5 | 1.15 | 281.2 | 105.7 | 116.7 | 107.3 | 1.109 | 0.656 | 0.926 | 0.897 |
| 0.25~0.50 | TS6 | B1 | 0.85 | 49.35 | 101.5 | 103.1 | 103.1 | 0.404 | 0.409 | 0.497 | 0.437 |
| | TS7 | B2 | 0.925 | 103.12 | 104.9 | 102.5 | 102.7 | 0.387 | 0.615 | 0.633 | 0.545 |
| | TS8 | B3 | 1.0 | 121.1 | 104.3 | 107.6 | 105 | 0.805 | 0.437 | 0.815 | 0.686 |
| | TS9 | B4 | 1.075 | 235.6 | 105.6 | 103.9 | 110 | 0.852 | 0.833 | 0.514 | 0.733 |
| | TS10 | B5 | 1.15 | 134.9 | 104 | 104.3 | 110 | 0.878 | 0.795 | 0.540 | 0.738 |
| 0.50~1.0 | TS11 | C1 | 0.85 | 43.15 | 103.9 | 101.5 | 101.7 | 0.515 | 0.609 | 0.633 | 0.586 |
| | TS12 | C2 | 0.925 | 91.45 | 102.6 | 104 | 103.4 | 0.579 | 0.658 | 0.743 | 0.660 |
| | TS13 | C3 | 1.0 | 109.1 | 107.6 | 103 | 103.8 | 0.711 | 0.711 | 0.609 | 0.677 |
| | TS14 | C4 | 1.075 | 162.3 | 104.4 | 109.7 | 104.5 | 0.753 | 0.792 | 0.702 | 0.749 |
| | TS15 | C5 | 1.15 | 208.2 | 106 | 107.4 | 106 | 0.612 | 0.649 | 0.628 | 0.630 |
| <0.25 | TS16 | D1 | 1.0 | 123 | 106 | 102.8 | 102.5 | 0.493 | 0.510 | 0.507 | 0.503 |
| | TS17 | D2 | 1.0 | 136.1 | 103 | 109.4 | 103.4 | 0.433 | 0.492 | 0.385 | 0.437 |

Analysing the ultrasonic wave velocity test results of the samples in Table 1, it can be seen that there is little difference in the ultrasonic speed in various directions, indicating that the prepared briquette samples have good homogeneity, providing support for the scientific rationality of subsequent analysis. In addition, as shown in Figure 5, by comparing the longitudinal wave velocities of different samples, the effects of particle size, density and moulding pressure on the test results can be clearly determined. In Figure 5a, as the density of the sample increases, the longitudinal wave velocity shows a rising trend, and the two show a positive correlation. This is also evident in the relationship between the forming pressure and the longitudinal wave velocity shown in Figure 5b. Obviously, the smaller the particle size, the higher the density and moulding pressure, the closer the bonding between particles, the faster the longitudinal wave velocity and the higher the linear correlation coefficient between parameters.

### 3.2. Natural Frequency Test Results

In the natural frequency test, the results of each test were not completely consistent because the parameters of hammering force, angle and duration are not easy to grasp. In view of this, the method of calculating the arithmetic mean of multiple hammer strikes was adopted to determine the natural frequency of the briquette specimen. Specifically, each side of the specimen in the triaxial direction (defined as the A-side, B-side and C-side) was tested at least three times, recording the force hammer excitation signals and acceleration response signals simultaneously and conducting targeted analysis of the above signals.

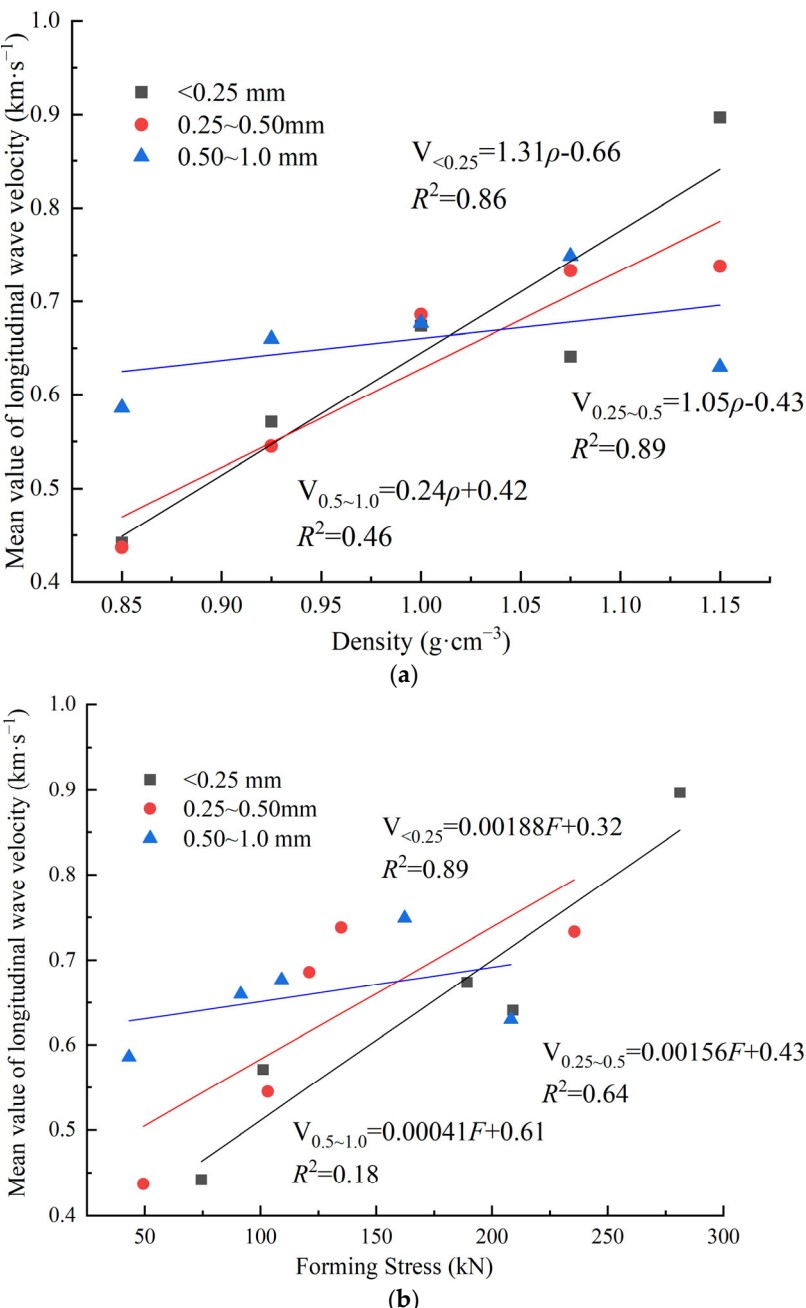

**Figure 5.** Longitudinal wave velocity of samples under different particle size, density and moulding pressure. (**a**) The relationship between wave velocity and density; (**b**) the relationship between wave velocity and pressure.

### 3.2.1. Excitation Signal Analysis

A large number of hammer excitation signals were obtained in the experiment. Due to space limitations, this article only selects some representative signals for discussion. Taking the first hammer test of the A-side of the C4 coal sample as an example, the excitation signal returned by the hammer is converted and amplified, as shown in Figure 6. As seen from Figure 6a, the duration of the hammer excitation is very short, only approximately 10 ms. Additionally, the signal contains significant noise information. By performing a fast Fourier transform (FFT), it was found that the coverage range of shock pulses is extremely wide. Similarly, from a frequency domain perspective, we can conclude that the signal is severely affected by noise interference, as shown in Figure 6b.

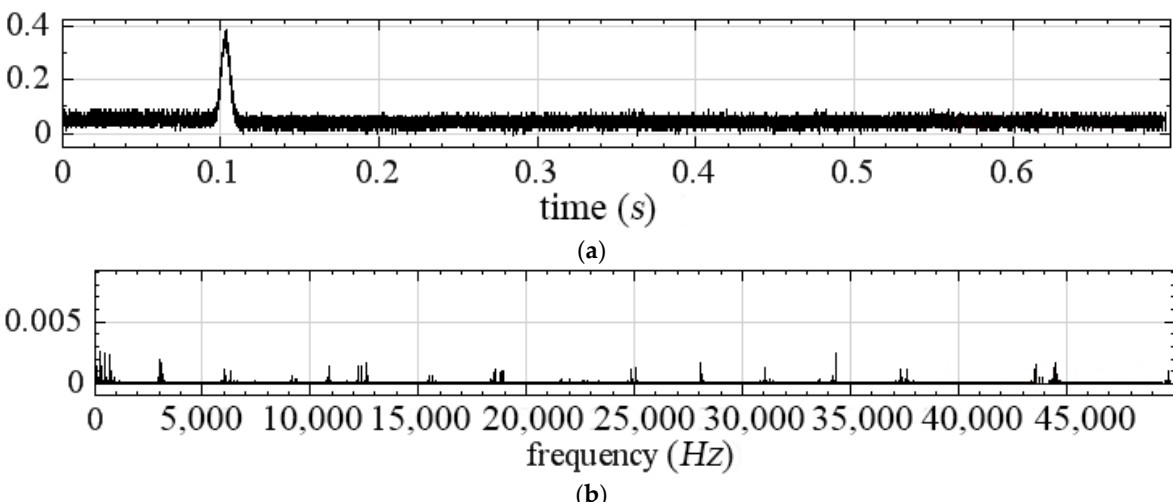

**Figure 6.** Time and frequency domain curves of typical excitation signals. (**a**) Time domain signal; (**b**) frequency domain signal.

Figure 6 shows that the excitation signal is a typical nonlinear, nonstationary transient signal, with the typical characteristics of high noise, a short duration and rapid mutation. Therefore, it can be denoised by the Hilbert–Huang transform (HHT) method [66,67]. The processing flow of HHT is shown in Figure 7. For the original signal X(t), all its maxima and minima values are first obtained, and then all local extreme points are interpolated using the cubic spline function to form the upper and lower envelopes of the signal, i.e., $X_{max}(t)$ and $X_{min}(t)$. Subsequently, the mean value of both, $X_{mid}(t)$, is calculated. The first component $h_1(t)$ is obtained by subtracting $X_{mid}(t)$ from the original data sequence X(t), and a decision is made as to whether or not $h_1(t)$ satisfies the conditions required for IMF. If they are satisfied, $h_1(t)$ is recorded as IMF1 and the modal decomposition is repeated for the X(t)–$h_1(t)$ data. If they are not satisfied, $h_1(t)$ is taken as the new original signal and the above steps are repeated until the data after several operations satisfy the screening conditions, which decomposes the first IMF signal from the original data [68]. Obviously, in the above process, the establishment of the screening termination condition is crucial, and this value can be determined by calculating the standard deviation of two consecutive screening results as a criterion. Experience has shown that when the termination condition is set at 0.2–0.3, it can ensure the linearity and stability of the IMF while also providing the IMF with corresponding physical significance.

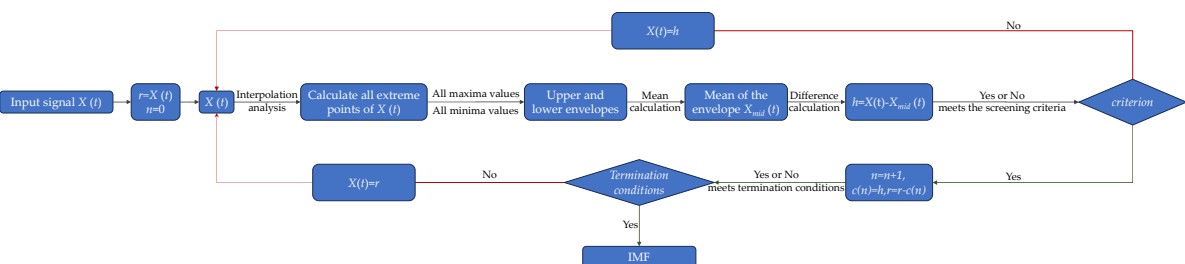

**Figure 7.** The program flow chart of HHT algorithm.

As shown in Figure 8a, the signal is decomposed into 15 intrinsic mode functions (IMFs) and 1 residual component. IMFs 1~7 do not exhibit significant pulse signal characteristics and have a low energy proportion, which is analysed to be the result of background noise interference.

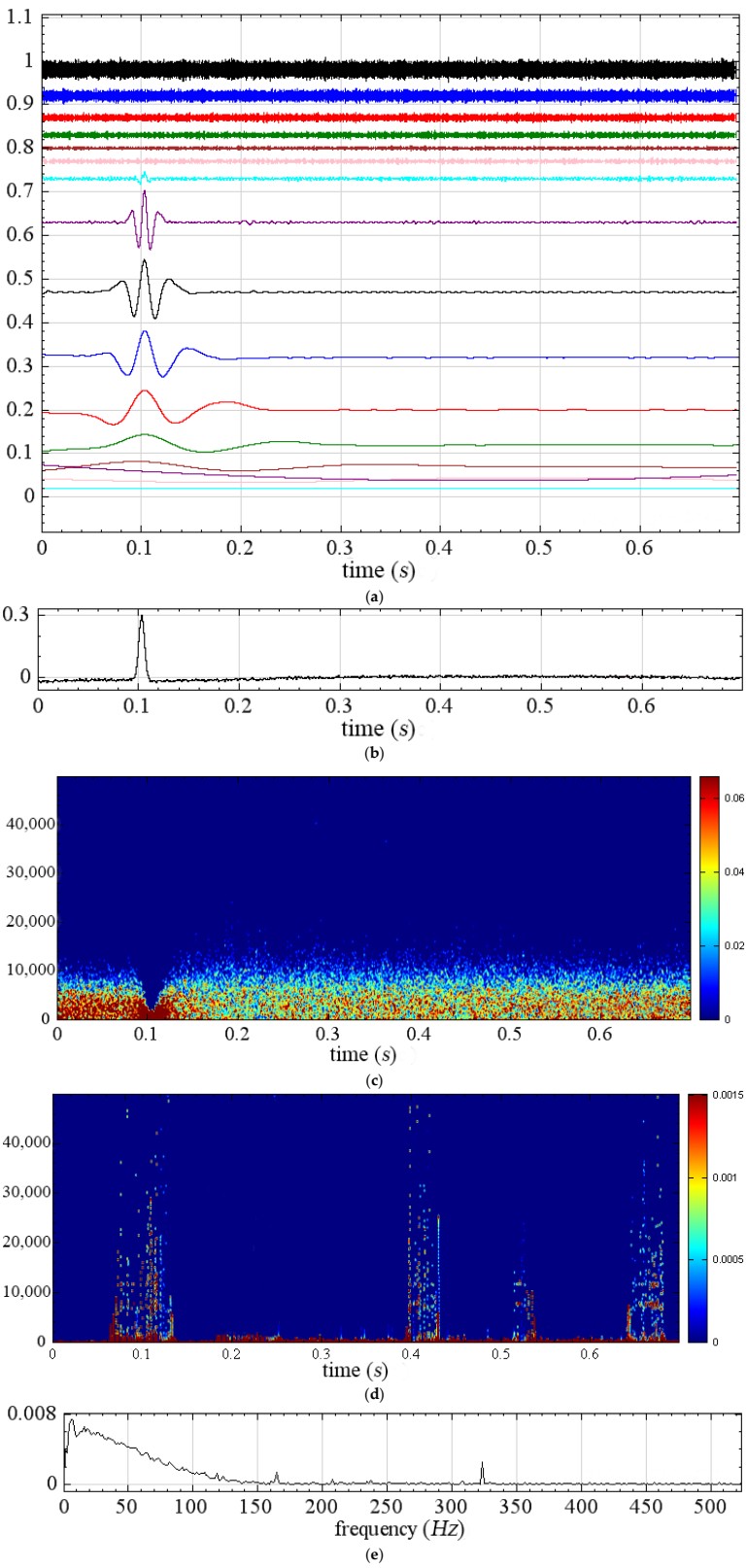

**Figure 8.** Typical denoising process of the excitation signal. (**a**) HHT decomposition of the original signal; (**b**) math function reconstruction of the effective components; (**c**) Hilbert spectrum of the original signal; (**d**) Hilbert spectrum of the reconstructed signal; (**e**) spectrum curve of the reconstructed signal.

The above background noise signals are eliminated, the effective signals are reconstructed by the math function and the resulting signal is shown in Figure 8b. Compared with Figure 6a, the signal amplitude has little significant change before and after denoising, but the background noise signal is effectively removed. To verify this, the Hilbert spectra of the excitation signals before and after denoising are compared, as shown in Figure 8c,d. The background noise, which is concentrated in the range of 50~1000 Hz, is effectively removed, while the effective signals are well preserved, and the signal-to-noise ratio of the signals is effectively improved.

In addition, an analysis of the frequency domain curve of the reconstructed signal (Figure 8e) shows that the effective excitation signal has a wide frequency band, covering a range of 0 to 500 Hz, with dominant frequency bands below 200 Hz and no obvious peaks. Therefore, the impulse excitation actually generates a wideband signal.

### 3.2.2. Response Signal Analysis

In the natural frequency measurement experiment, a large number of response signals were also obtained. Taking the second hammering test on the A-side of the C1 coal sample as an example, the time domain curve of the original signal is shown in Figure 9a. Obviously, this signal is also interfered with by external noise. Referring to the denoising method of the excitation signal, the HHT method is applied for IMF decomposition and effective signal math reconstruction. The obtained effective signal is shown in Figure 9b. By performing an FFT, it is known that the dominant frequency of the signal is 33.07 Hz, as shown in Figure 9c.

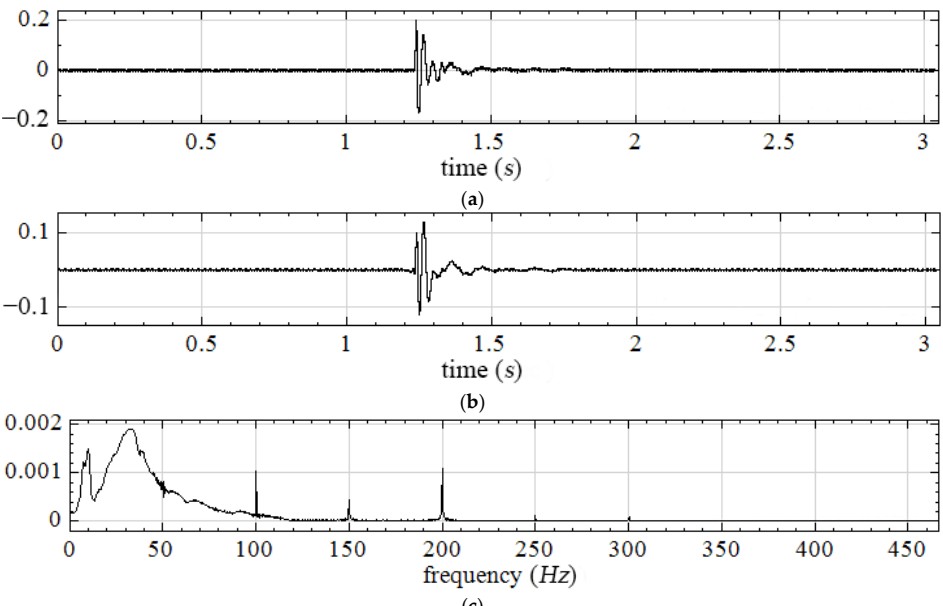

**Figure 9.** Typical denoising process of the response signal. (**a**) Original signal; (**b**) reconstructed signal; (**c**) spectrum curve of the reconstructed signal.

The above method is applied to remove noise from the response signals, record the dominant frequencies of each signal and calculate their arithmetic mean values. The final results are shown in Table 2. Table 2 indicates that there is no significant difference between the final calculated frequencies for each test, which reflects the accuracy of the test results and denoising analysis. Notably, some test results show the "false frequency phenomenon", indicating a high degree of dispersion in the measured natural frequency, with values less than 10 Hz. This may be due to improper operation. To facilitate the analysis, the above signals are presented in the form of "---".

**Table 2.** Natural frequency test results of the briquettes.

| Particle Size Grade (mm) | Group ID | Test Face ID | Dominant Frequency after Denoising (Hz) | | | Arithmetic Mean of Each Test Surface (Hz) | Sample Natural Frequency (Hz) |
|---|---|---|---|---|---|---|---|
| <0.25 | A1 | A | 32.43 | 33.83 | 33.13 | 33.13 | |
| | | B | 28.90 | 42.67 | 29.01 | 33.53 | 34.27 |
| | | C | 39.40 | 33.90 | 35.20 | 36.17 | |
| | A2 | A | 40.37 | 44.09 | 42.34 | 42.27 | |
| | | B | 43.24 | 44.99 | 39.95 | 42.73 | 43.34 |
| | | C | 44.80 | 47.59 | 42.67 | 45.02 | |
| | A3 | A | 19.91 | 26.91 | 28.34 | 25.06 | |
| | | B | 46.22 | 39.11 | 40.34 | 41.89 | 34.65 |
| | | C | 36.62 | 36.62 | 37.74 | 37.00 | |
| | A4 | A | 39.25 | 39.40 | 41.48 | 40.05 | |
| | | B | 19.86 | 19.73 | 20.44 | 20.01 | 32.71 |
| | | C | 37.15 | 37.18 | 39.91 | 38.08 | |
| | A5 | A | 32.79 | 31.91 | 30.51 | 31.73 | |
| | | B | 39.52 | 38.18 | 38.40 | 38.70 | 36.95 |
| | | C | 40.53 | 40.75 | 39.92 | 40.40 | |
| 0.25~0.50 | B1 | A | 36.69 | 35.91 | 38.40 | 37.00 | |
| | | B | 27.24 | 29.54 | 27.73 | 28.17 | 36.28 |
| | | C | 43.20 | 47.29 | 40.53 | 43.67 | |
| | B2 | A | 34.61 | 32.53 | 33.28 | 33.47 | |
| | | B | 33.66 | 34.52 | 32.85 | 33.68 | 33.80 |
| | | C | 35.05 | 33.22 | 34.46 | 34.24 | |
| | B3 | A | 34.74 | 34.99 | 32.82 | 34.18 | |
| | | B | 34.99 | 33.75 | 35.41 | 34.72 | 35.59 |
| | | C | 38.79 | 37.18 | 37.62 | 37.86 | |
| | B4 | A | 35.68 | 33.78 | 33.83 | 34.43 | |
| | | B | 29.87 | 29.07 | 32.00 | 30.31 | 32.59 |
| | | C | 32.85 | 32.71 | 33.52 | 33.03 | |
| | B5 | A | 33.36 | 33.42 | 31.29 | 32.69 | |
| | | B | 29.56 | 29.16 | 28.88 | 29.20 | 30.79 |
| | | C | 28.92 | 28.88 | 33.60 | 30.47 | |
| 0.50~1.0 | C1 | A | 32.49 | 33.07 | 35.91 | 33.82 | |
| | | B | 36.46 | 36.98 | 34.91 | 36.12 | 35.38 |
| | | C | 36.27 | 36.85 | 35.45 | 36.19 | |
| | C2 | A | 36.98 | 36.27 | 37.97 | 37.07 | |
| | | B | 41.60 | 40.89 | 35.35 | 39.28 | 37.65 |
| | | C | 33.85 | 37.83 | 38.13 | 36.60 | |
| | C3 | A | 38.83 | 35.20 | 37.69 | 37.24 | |
| | | B | 34.99 | 35.30 | 36.46 | 35.58 | 36.42 |
| | | C | 37.12 | 35.12 | 37.12 | 36.45 | |
| | C4 | A | 34.99 | 34.99 | 36.03 | 35.33 | |
| | | B | 33.66 | 33.66 | 34.67 | 34.00 | 36.24 |
| | | C | 40.73 | 38.04 | -- | 39.39 | |
| | C5 | A | -- | -- | -- | -- | |
| | | B | 32.97 | 34.13 | 32.71 | 33.27 | 34.49 |
| | | C | 36.10 | 35.12 | 35.91 | 35.71 | |

The results shown in Table 2 are plotted as a scatter plot of natural frequencies, as shown in Figure 10. It can be seen that the natural frequencies of the coal samples are located between 30.79 Hz and 43.34 Hz, with most of them fluctuating around 35 Hz. This conclusion provides a basis for the subsequent setting of the dynamic load frequency.

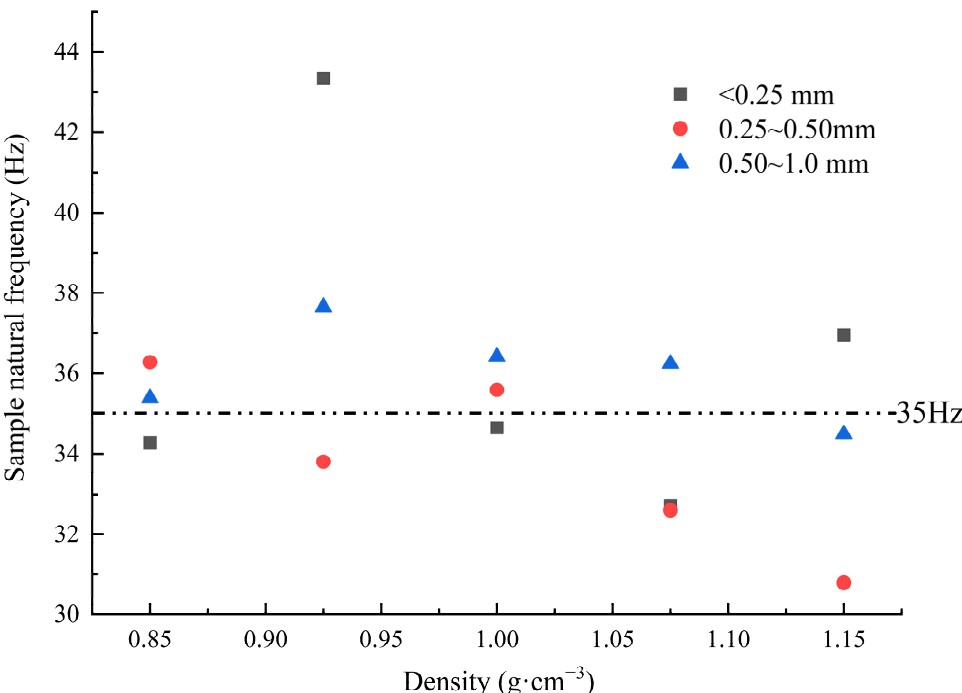

**Figure 10.** Scatter plot of coal natural vibration frequencies.

### 3.3. Characteristics of MS Signals during the Static Load Failure Process

3.3.1. Evolution Law of the MS Signal Amplitude and Spectrum

As mentioned earlier, in the process of "critical static stress + slight disturbance"-induced rockburst, whether the critical state has been formed and can continue to be maintained is the key prerequisite for the instability of the coal body. As the critical state is formed under the action of static stress, samples D1 and D2 in Table 1 are placed in the clamping subsystem (Figure 3a). The static load subsystem is started, the manual hydraulic pump is used to slowly pressurize the system and the load is uniformly transferred to the test piece through the jack, resulting in uniaxial compression of the sample. A precision digital pressure gauge (10,000-point scale) is attached to the top of the hydraulic pump. During the experiment, the readings of the pressure gauge are recorded synchronously until the pressure gauge reading suddenly drops from a high level to a large amplitude, indicating that the coal body has lost its load-bearing capacity and that the static load damage experiment is complete. In this process, the MS sensor is coupled to the side surface of the coal body by using petroleum jelly, and the ZDKT-1 data acquisition system is turned on at the same time. The sampling frequency is set to 5120 Hz, and a data file is formed every 10 s on average to record the MS signals during the experiment.

Since the patterns exhibited by the D1 and D2 specimens are basically the same, limited to space, only D1 is used as an example for illustration. By testing, sample D1 experienced a total of 670 s from the beginning of loading to the final destabilization. During this process, a large number of MS signals were recorded. The typical signals among them are listed as shown in Figure 11.

Figure 11a shows the MS signals captured in the first 10 s of the experiment, indicating that the signal amplitude is small and lacks regularity. The spectrum is analysed using FFT, as shown in Figure 11b. The dominant frequency of the signal is found to be 50 Hz, and its frequency-doubled signals and white noise belong to the typical alternating current (AC) interference and white noise within the monitoring system [69] and do not contain effective signals.

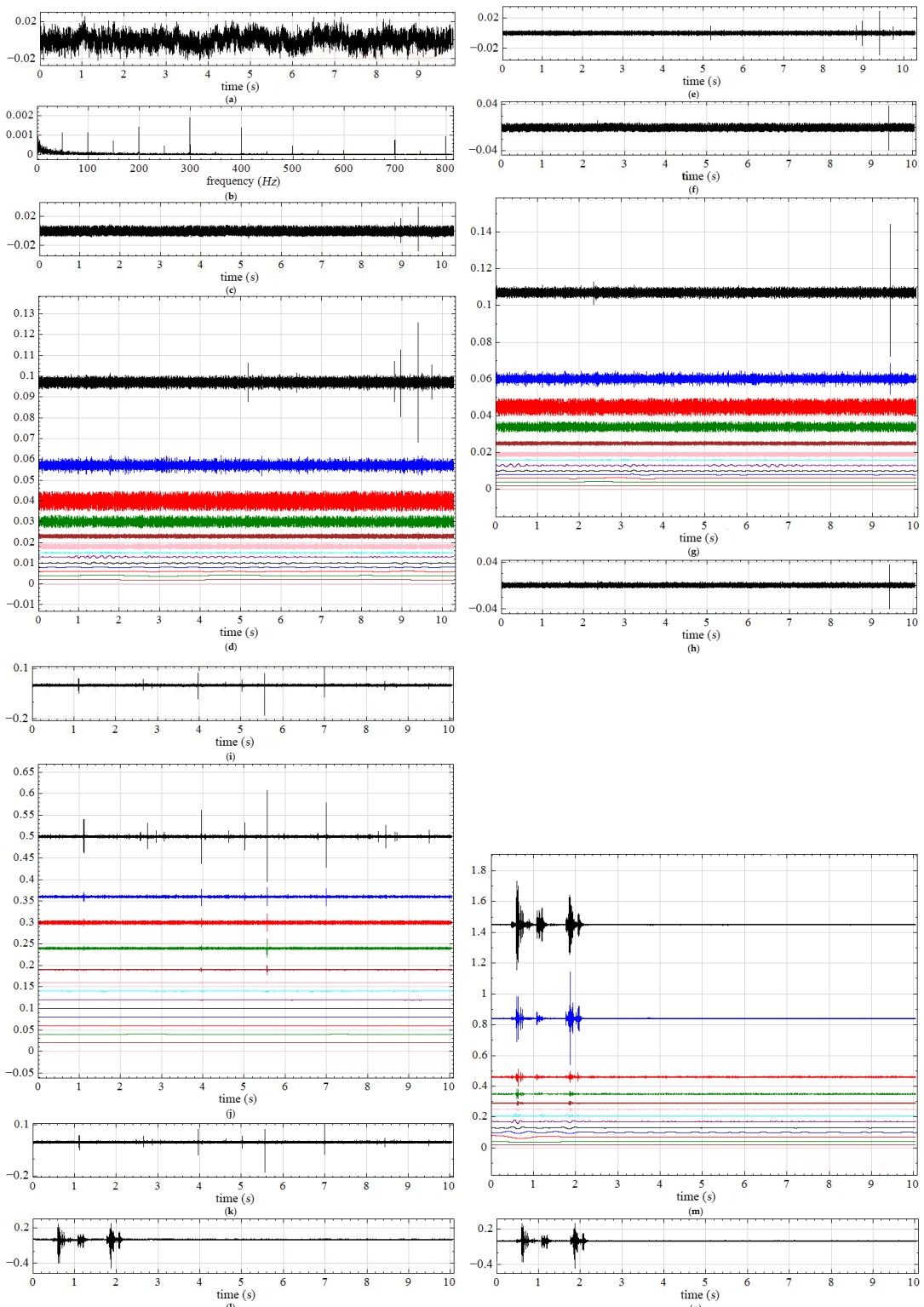

**Figure 11.** Characteristics of MS signals during the static load failure process of a coal briquette. (**a**) Original signal for 0–10 s. (**b**) Spectrum analysis of the 0–10 s signal. (**c**) Original signal for 10–20 s. (**d**) HHT decomposition of the 10–20 s signal. (**e**) Reconstruction of the math function for the 10–20 s signal. (**f**) Original signal for the 100–110 s. (**g**) HHT decomposition of the 100–110 s signal. (**h**) Reconstruction of the math function for the 100–110 s signal. (**i**) Original signal for 280–290 s. (**j**) HHT decomposition of the 280–290 s signal. (**k**) Reconstruction of the math function for the 280–290 s signal. (**l**) Original signal for 510–520 s. (**m**) HHT decomposition of the 510–520 s signal. (**n**) Reconstruction of the math function for 510–520 s signals.

As static press loading progressed, the monitoring system recorded effective MS signals for the first time after the experiment was carried out for 15~20 s, as shown in Figure 11c. Due to the distinct temporal nonlinear characteristics of the MS signal during the process of coal static damage, the HHT method can still be used to decompose the signal. As shown in Figure 11d, the signal is decomposed into 13 IMF components and 1 residual component, and only the IMF1 component has typical MS signal characteristics, while the remainder of the components are noise signals, so the IMF1 component can be extracted as a valid signal (Figure 11e).

Similarly, Figure 11f,g reflect the MS signals and their HHT decomposition results when the experiment was conducted for 100 s. Compared with Figure 11c,d, on the one hand, the number of MS signals gradually decreases, while the amplitude starts to increase; on the other hand, the number of effective IMF components begins to increase, indicating a reduction in the dominant frequency of MS signals. Obviously, both IMF1 and IMF2 of the 100~110 s signals contain valid information, so both of them are subjected to MATH reconstruction, and the calculated results are used as the final valid MS signals, as shown in Figure 11h.

Continuing loading at the above rate, we observed that the frequency of valid signals became more intensive, and a transition occurred from a "single signal" to "dual signals" to even "group signals". The typical signals are shown in Figure 11i, which are recorded by the monitoring system in 280–290 s. From the figure, we monitored at least seven valid signals in 10 s. Not only is the amplitude of the valid signals larger than before, but the number of IMFs containing valid information after decomposition also reaches a record number of five for the first time (Figure 11j). Using the Math function to reconstruct the above IMFs, the effective signal obtained is shown in Figure 11k.

The same process is also illustrated in Figure 11l. Notably, Figure 11l reflects the original signals at 510–520 s. Compared with Figure 11i, there are both similarities and significant differences. The same point is that both are group signals. In fact, after 290 s, except for a few periods, most of the effective MS signals belong to this category. The differences are mainly reflected in the following aspects: (1) As the loading time progresses, the amplitude of the MS signals gradually increases, indicating that the destruction process of the coal body is accelerating synchronously, and the physical and mechanical state of the coal body is deteriorating. (2) The effective components extracted from the HHT decomposition in Figure 11j are IMF1 to IMF5, while the effective components in Figure 11m are IMF1~IMF7, which indicates that the dominant frequency of the MS signal is still declining. (3) Compared to Figure 11i, the group signals shown in Figure 11n appear more intense, and the signals appear in a "cluster" pattern. In fact, the continuous occurrence of more than three large amplitude MS signals and multiple clusters of dense signals is the most prominent feature of the recorded signals in the 510–520 s interval.

### 3.3.2. The Correlation between MS Energy and Static Pressure Load

For comparison and analysis, the effective signal energy in each time period is calculated at 10 s intervals on the basis of MS signal denoising and reconstruction, and the final results are shown in Figure 12. In the figure, we also added manually recorded pressure gauge readings at typical time points during the experiment. In fact, the effective MS signals mentioned in the previous section are encapsulated in Figure 12. Point (a) shows the background noise recorded by the monitoring system during the start-up of the static system. At this moment, although the manual hydraulic pump has been used to slowly pressurize the system, the static load has not yet been applied to the tested piece though the jack. The monitored signal was actually the result of interference from internal and external noise sources in the acquisition system. Point (b) represents the first group of MS signals detected by the test system. After denoising analysis, it is found that although the effective signal contains typical MS signal characteristics, its amplitude and energy are low, and the number of MS events included in the data is also very low. The corresponding digital pressure gauge reading of the jack at this time is 1.548 MPa, indicating that the coal

body is in an initial damage state, and no surface cracks can be observed by the naked eye. In the following one-minute period, the signals recorded by the monitoring system are mostly background noise, with the appearance of occasional small-amplitude MS signals. The corresponding pressure readings increased to 2.307 MPa and then fluctuated in a small range, without any significant increase in value. These results indicate that the briquette specimen is in the early stage of nonlinear compression and densification (stage I). The sample does not exhibit large lateral expansion, and the overall volume decreases with increasing load.

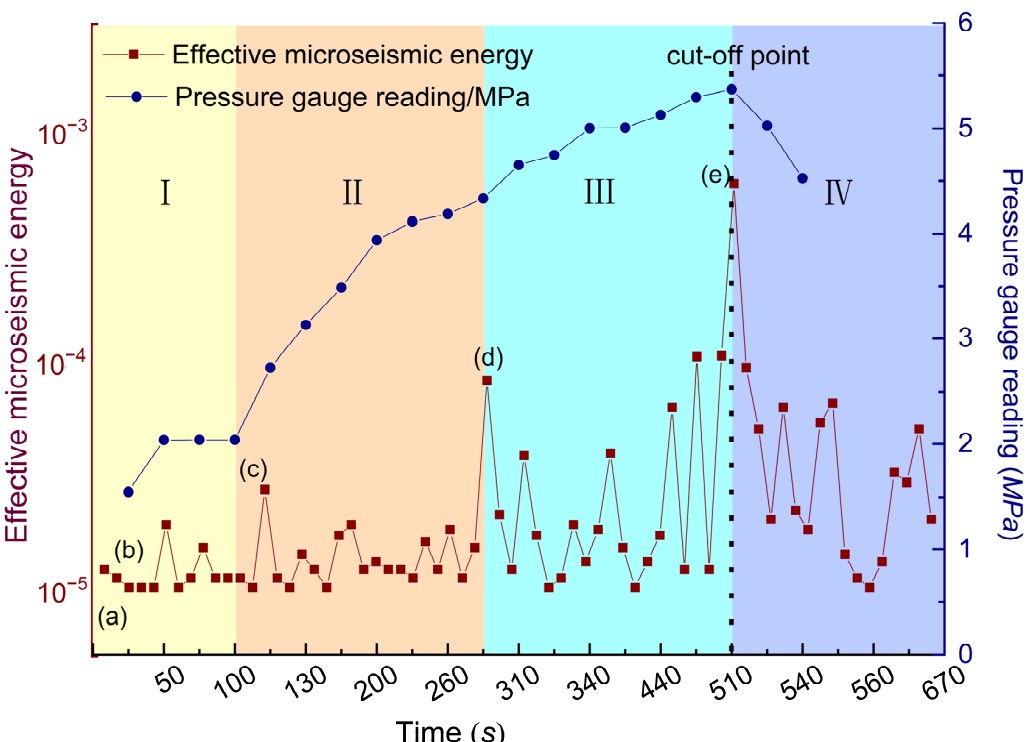

**Figure 12.** Correlation between the MS energy and pressure gauge readings during the static load failure process. (I) Early stage of nonlinear compression and densification; (II) development stage of microcracks; (III) nonstable stage of microrupture; (IV) stage of losing load-bearing capacity; (a) static load system startup; (b) the first group of MS signals appear; (c) generation of MS signals during the linear elastic stage; (d) effective signal peak time in the fully elastic stage; (e) effective signal peak time throughout the entire process.

When the static pressure is loaded for 100 s, on the one hand, the number of MS signals begins to decrease until there is only one effective signal in each 10 s monitoring period, but their amplitude begins to increase and the energy of the signals begins to rise slowly. On the other hand, the dominant frequency of the signals shows a decreasing trend, indicating that the briquette is in stage II, i.e., although no macroscopic cracking is observed, the microcracks within the specimen are steadily developing. At the same time, the reading of the pressure gauge increases linearly, indicating that the sample is undergoing a linear elastic deformation stage. Maintaining the above loading rate, we find that the effective signal energy suddenly reaches a peak at approximately 280 s. The MS original signal in Figure 11i is the result of a large number of high-amplitude signals being generated densely. At this point, the reading of the pressure gauge reached 4.330 MPa.

In the following experimental process, on the one hand, the intensity of the MS signals began to intensify, and the signal amplitude also increased rapidly. On the other hand, the upward trend of the pressure gauge readings began to slow down, and the "crackling" sound caused by the damage of the coal body was heard during the experiment. This indicates that the sample has transitioned from a completely elastic body to an elastic–

plastic state and that the microrupture has entered the nonstable stage (stage III). The specimen changes from the previous volume compression to expansion, and the axial deformation increases rapidly.

During the subsequent experiment, not only did the amplitude and energy of the signal increase synchronously, but the surface fracture phenomenon was observed on the surface of the coal body for the first time. Until the test was carried out for 510 s, more than three large amplitude MS signals and multiple clusters of signals appeared consecutively. After that, when the hydraulic pump was slowly applied to the sample, the growth trend of the pressure gauge reading started to slow down or even stopped changing. However, the evolution speed of the cracks on the surface of the coal body began to accelerate. Within 10 s, the cracks developed rapidly, crossed and united with each other to form a macroscopic fracture surface. The pressure gauge reading rapidly decreased to 4.525 MPa, indicating that the specimen had been damaged and lost its load-bearing capacity.

Figure 12 shows that there is a certain correlation between the energy of the effective MS signals and the pressure gauge reading. At point (e), the value of the signal energy in the 10 s time period reaches its maximum, and the pressure gauge reading also reaches the extreme value at the same time. Thereafter, if the loading is continued, the deformation of the briquette is mainly reflected in the sliding of the blocks along the macroscopic fracture surface. The sample lost its load-bearing capacity, and the reading of the jack rapidly decreased.

During the static loading damage test, when there are more than three large amplitude MS signals and multiple clusters of dense signals occur consecutively, the load-bearing stress of the briquette simultaneously reaches a peak value. At this time, although the sample has not been completely destroyed, it has entered the critical state and will lose the load-bearing capacity immediately in the subsequent static loading. Based on the above analyses, we propose that the continuous cluster of three or more large amplitude MS signals is an important criterion for the critical damage of briquettes under static loading, and we conducted the subsequent static–dynamic combination loading tests based on this. It should be noted that this conclusion is obtained from the testing and analysis of the briquette sample from the Tashan Mine and that its general applicability remains to be verified.

*3.4. Characteristics of MS Signals under Weak-Energy and Low-Frequency Disturbances*

3.4.1. Extraction of MS Signals in a High-Noise Background

As mentioned earlier, the destruction process of the coal body is often accompanied by the occurrence of a large number of MS signals. Therefore, similar to the static load process analysis, we use MS signal characteristics to reflect the deformation and structural instability process of coal under the coupling of static load and weak-energy and low-frequency dynamic load. The specific experimental procedure is shown in Figure 4 in the previous section, in which the briquette sample is first placed in the clamping subsystem, and the static loading test is carried out using the manual hydraulic pump. When more than three large amplitudes and multiple clusters of dense MS signals appear consecutively, the jack pressure is immediately reduced by 20% and maintained. Subsequently, the combined dynamic and static loading test is carried out. The input frequency and amplitude of the signal generator are adjusted to generate sinusoidal excitation. To facilitate the analysis, the excitation frequency of the dynamic loading is set at 35 Hz for each sample, with reference to the test results of the natural frequency. The test is conducted in the loading mode of "1 h of excitation—1 h of pause—1 h of excitation", and the MS signals are recorded simultaneously during the test.

The monitoring system records a large number of MS signals. Due to space limitations, the following example illustrates the signals of the second vibration of the B2 coal sample during 3560–3570 s. The original signal is shown in Figure 13a. In the initial identification of Figure 13a, it is determined that there are significant signals within this time period but also a significant amount of background noise. Initially, we refer to the HHT decomposition method

used in the static load denoising process and find that the IMF 1~IMF 3 components contain valid information. However, the IMF 1~IMF 3 components are also heavily affected by noise, i.e., the phenomenon of overlapping the frequency bands of the valid and background signals occurs. If the Math reconstruction is performed directly, it will be difficult to achieve effective signal denoising. FFT analysis of the original signal reveals that interference noise signals mainly include two categories: (1) nonstationary frequency band noise, which contains random white noise and some other noises (Part I in Figure 13c); and (2) fixed-band noise (Part II in Figure 13c). In addition to interference from 50 Hz and its multiples, the fixed-band noise is more dominated by 35 Hz and its multiples of the exciter carrier signal. Significantly different from static pressure loading, the amplitude and intensity of the carrier signal are much larger than those of the industrial frequency, even causing some effective signals to be buried in it. In view of this, we adopt the method of digital filtering and EMD noise reduction for signal processing, i.e., first, the 35 Hz, 50 Hz and their octave band frequencies are filtered and preprocessed, and then the processed EMD decomposition and reconstruction are performed by using the HHT. The final effective signals are obtained as shown in Figure 13d. Comparing Figure 13a,d, it can be seen that due to the influence of "powerful" noise, some effective signals are indeed submerged in it. The method of digital filtering superimposed with HHT noise reduction can be used to extract weak signals against the background of high noise and a low signal-to-noise ratio, with good application results.

### 3.4.2. Energy Statistics of MS Signals

Using the analysis method in the previous section, digital filtering and HHT noise reduction are applied to all MS signals of the B2 briquette. To facilitate comparison, the effective MS signal energy is calculated and time-averaged during each monitoring period, and the final MS signal energy distribution characteristics obtained are shown in Figure 14. From the figure, it can be seen that the energy density of the effective MS signal reaches the maximum value at the instant of exciter activation. The reason for this may be that a large number of microfractures have been formed in the static loading process of the coal. The weak-energy and low-frequency dynamic disturbance accelerates the opening and closing of the microcracks while causing the rapid expansion of the crack tip. The microfractures converge, gather and penetrate to form a main fissure. These factors result in the intensive occurrence of high-amplitude effective MS signals during this time period, with signal energy reaching a peak value. In the subsequent experimental process, the specimen rupture entered the bottleneck stage, and the monitored MS energy continued to remain at a low level of oscillation. Even the high-energy MS signals generated by chance are mostly single signals, and this phenomenon continues until the end of the first dynamic loading stage (1 h).

In the second dynamic loading stage, the energy of the MS signals remains low in the beginning, but its value climbs rapidly after 6000 s. The signals appear intensively at this time. The reason for the above phenomenon is that the weak-energy and low-frequency disturbance essentially produces a kind of fatigue failure. There are significant differences in the damage modes caused by fatigue, static loads and impact dynamic loads. As the number of perturbations increases, the sample cracks expand continuously, and the stress threshold for failure instability also decreases. Referring to the fatigue damage law of metal specimens, the failure of briquettes actually undergoes three stages: the near-threshold stage, the rapid propagation stage (the Paris zone) and the final fracture stage. In the initial stage, the expansion rate of fatigue cracks is very small. More importantly, the initial static load in the experiment is actually an overload, which causes a large plastic zone to form at the crack tip, resulting in a hysteresis effect of crack extension. When the subsequent static load is kept at 80% of the initial stress, the existence of residual compressive stress also slows down fatigue crack expansion [70]. In the second stage, referring to the Paris formula [71], the crack propagation rate is basically constant, and its value is mainly controlled by the amplitude of the stress field intensity factor at the crack tip. As the value of the crack tip stress field strength factor amplitude continues to increase, the crack expansion progresses

to the final fracture stage (stage III). In this stage, the crack expansion rate increases rapidly and destabilizes the specimen in a short time.

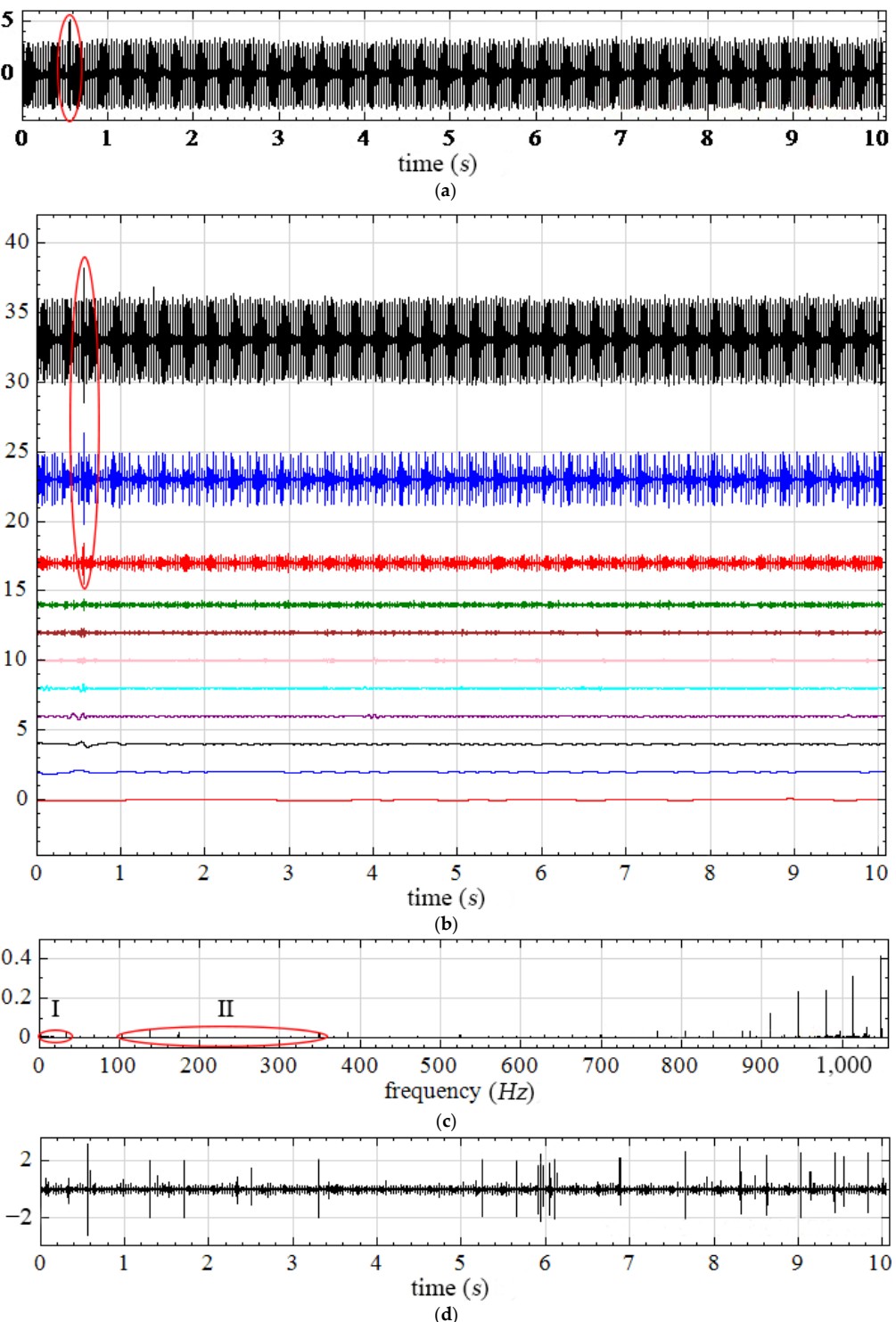

**Figure 13.** Signal denoising analysis under a strong noise background. (**a**) Original signal; (**b**) HHT decomposition of signals; (**c**) frequency domain curve of signal; (**d**) reconstructed signal.

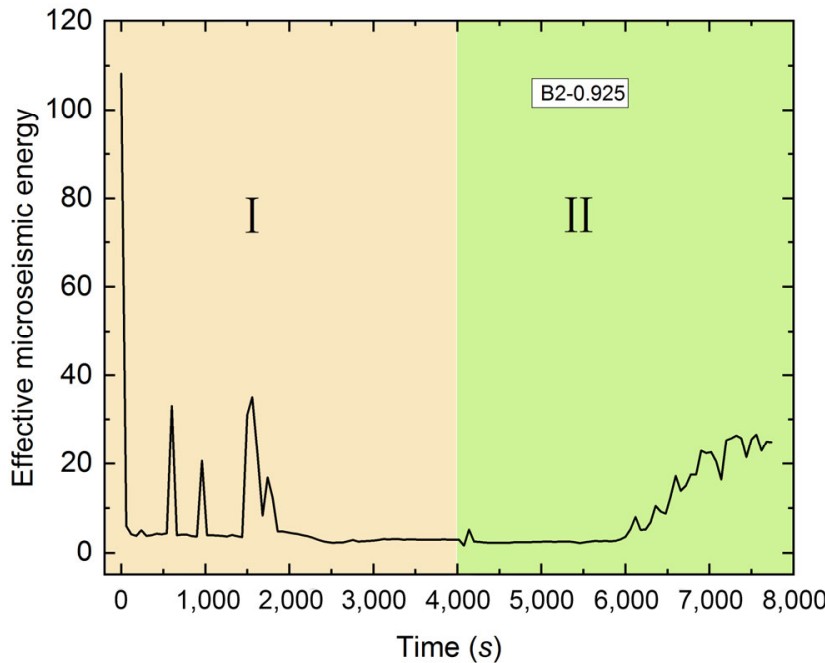

**Figure 14.** Time-history curve of the effective MS signal energy distribution (B2 sample). (I) First dynamic loading stage; (II) second dynamic loading stage.

Using the above method, the effective MS signal energies of all the coal samples are summarized and plotted as shown in Figure 15. Although the data have a certain dispersion, the following basic rules can still be reflected from the figure.

For the effective MS signals of the <0.25 mm group coals under weak-energy and low-frequency perturbation conditions, the MS energy of the secondary dynamics is much greater than that of the first dynamic loading (Figure 15a). This difference becomes less obvious in the 0.25~0.5 mm group. When the analysed object is the 0.50~1.0 mm group, there is even a trend where the energy of the first dynamic load is often greater than that of the second dynamic load, indicating that the evolution process of cracks is significantly advanced. In addition, a comparison of the time-course curves of the MS signals of coals under the same moulding pressure and at different constituent particle sizes (such as A1, B1 and C1) in Figure 15a~c, combined with the differences in the physical properties of the coal specimens, indicates that the smaller the particle size is, the higher the density of the samples. This results in a stronger effective MS signal energy generated during the failure process of the briquette, a longer duration of crack propagation from the near-threshold stage to the rapid propagation stage and a stronger ability of the coal sample to resist weak-energy disturbances.

In general, when the composition of the particle size is the same, the higher the moulding pressure is, the higher the densification degree of the specimen, and the more uneven the distribution of the effective MS signal time curve formed. Specifically, when the forming pressure is low, the composition of the specimen is relatively loose, and the initiation, propagation and penetration of microcracks often occur at all times, with the energy curve showing a low level of small-amplitude oscillation. In contrast, the higher the forming pressure is, the greater the difficulty of crack formation and the higher the amplitude and energy of the MS signals recorded on a single occasion, and the nonuniformity and abruptness of the energy curve are more obvious.

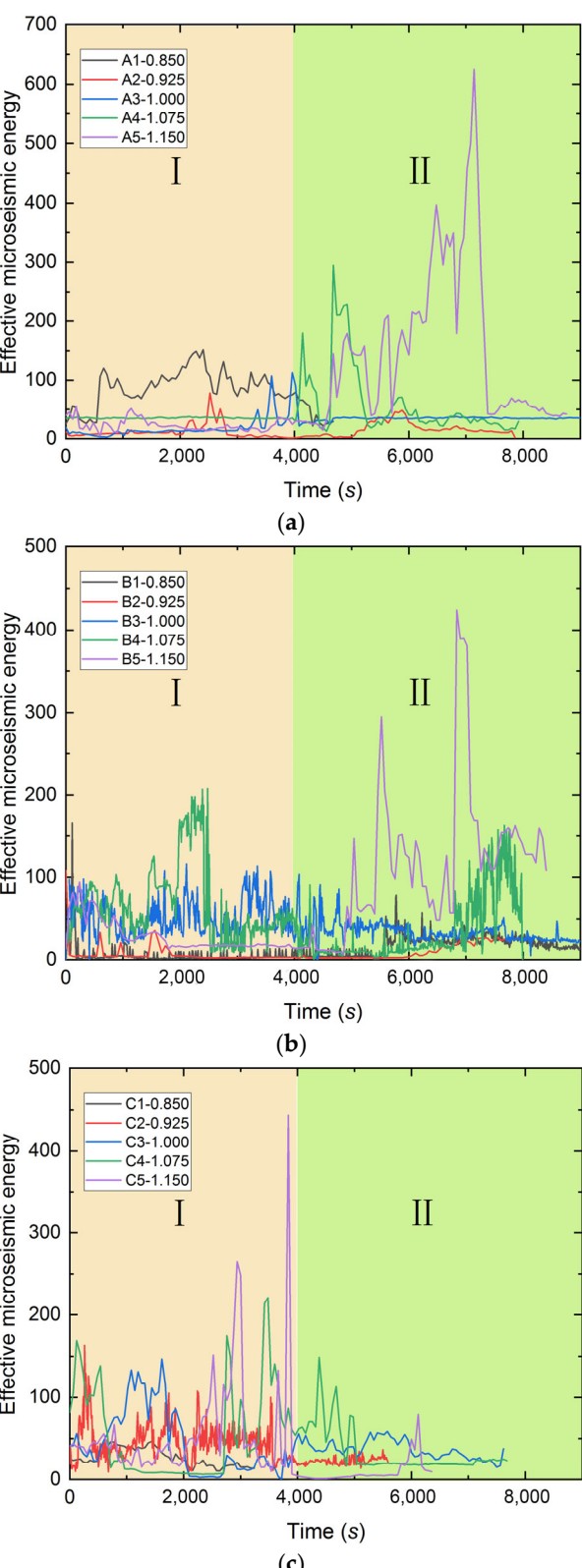

**Figure 15.** Summary of the MS effective signal energy under weak-energy and low-frequency disturbances. (**a**) Briquette with particle size < 0.25 mm; (**b**) briquette with particle size ranging from 0.25 mm to 0.50 mm; (**c**) briquette with particle size ranging from 0.50 mm to 1.00 mm.

## 4. Discussion

The induced modes of coal and rock destabilization and rockburst include two combined dynamic and static loading methods, i.e., elastic static stress + impact disturbance and critical static stress + slight disturbance. Previous studies have shown that, compared with the shallow coal layer, the occurrence mechanism of deep coal and rock dynamic disasters is closely related to the specific stress environment, i.e., high geostress and external disturbances [24]. The destabilization of the coal-rock system is an important prerequisite for coal and rock dynamic disasters, and disasters often go through the four stages of incubation, activation, development and termination [72]. The deep coal and rock mass is typically under a dynamic–static combination stress state. During the initial mechanical incubation stage, the coal and rock mass is in a three-dimensional stress equilibrium state under the influence of high static stress (point A in Figure 16). As some coal and rock bodies are extracted, the mechanical equilibrium state is broken, loads are transferred, fractures begin to evolve and damage occurs in the coal-rock system. In most cases, the changes in the abovementioned physical and mechanical parameters and state properties are insufficient to reach the conditions for the instability and failure of coal and rock. At this time, although the coal and rock system of the working face has experienced initial damage, it has not yet reached the point of massive destruction and structural instability, and it still maintains a certain "critical state" tenaciously, accumulating more potential energy [73]. Since the coal and rock mass is triggered at any moment, when it is affected by external dynamic disturbances, it is possible to quickly destroy and lose stability, leading to the occurrence of coal and rock dynamic disasters. Statistics show that 93.3% of rockburst behaviours are induced by external dynamic disturbance [74]. Thus, the external dynamic disturbance plays a crucial role in the coal and rock dynamic disaster from the mechanical gestation state to the rapid excitation stage [33,75], and targeted research on this issue is more meaningful.

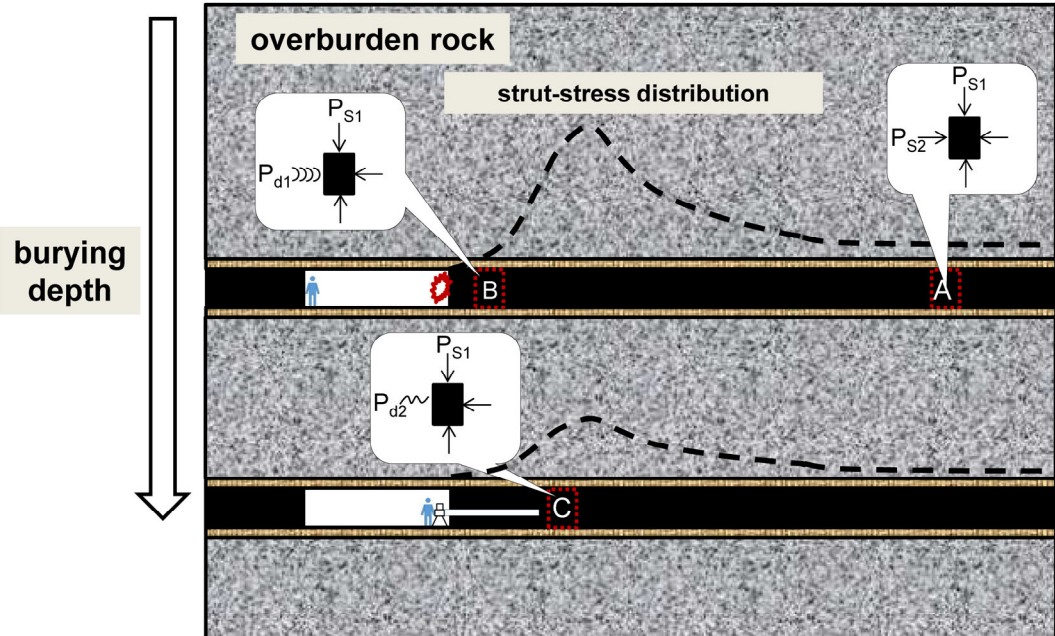

**Figure 16.** Schematic diagram of rockburst induced by different dynamic disturbances.

A large number of external dynamic disturbances exist in coal mines, such as tunnelling and blasting, coal caving, drilling construction, support finishing, mechanical vibration and even disturbances in nearby areas. These disturbances can be divided into two categories, impact disturbances (strong disturbances) and slight disturbances (weak disturbances), based on the energy level, load action form and magnitude of impact on

the change in coal and rock states [76]. Compared with strong disturbances, weak-energy disturbances mainly involve drilling construction, support repair, the vibration of large equipment and other forms of operation with relatively low energy, a long duration and distinct periodic characteristics. Their impact on the stability of coal-rock systems appears to be small and often easy to ignore, and experts worldwide have paid less attention to weak-energy disturbances. However, when the distance from the operation point is far away, the shock wave with a steep wavefront transforms into a seismic wave dominated by simple harmonic and elastic effects [35], i.e., the strong disturbance gradually decays to a weak disturbance. More importantly, in the process of mining, strong disturbances such as blasting and rock cross-cut coal uncovering are usually operated at a long distance, and there are complete safety measures before and after the operation, allowing personnel to evacuate relatively safely even in the event of an accident (point B in Figure 16). For close-contact operations such as drilling and support repair, a large number of people gather in the excavation space; most of them are in a state of low awareness of prevention, and the consequences of an accident are often more serious (point C in Figure 16). Therefore, compared with strong disturbances, weak-energy disturbance-induced rockbursts have a certain degree of concealment, and their threat to the workforce is also greater.

We believe that different stress conditions will lead to different damage patterns. Among the two major power disturbances mentioned above, there are not only differences in energy levels between impact disturbance and weak-energy disturbance but also many other differences as follows: (1) The degree of influence on stress distribution is different. A strong disturbance (such as coal cutting and blasting) comes with excavation footage, so its impact on the stress distribution of the coal-bearing body is comprehensive. It can make a wide range of coal body horizontal positive stress disappear, directly transforming the coal body from a three-dimensional stress state to a two-dimensional or even one-dimensional stress state. Furthermore, a strong disturbance can also cause a sudden increase in vertical superimposed stress, making the coal body in the aforementioned unfavourable boundary state more susceptible to damage and instability. Obviously, a weak-energy disturbance that does not come with a wide range of footage does not possess this characteristic, and its change in stress distribution is relatively small (Figure 16). (2) The path to damage the coal structure is different. A strong disturbance causes direct impact damage to the coal body, while a weak-energy disturbance is in the form of a vibration wave to the coal body reciprocal loading and unloading effect, representing fatigue damage to the coal body. (3) The prerequisite for coal body instability is different. A strong disturbance has high energy levels and can directly bring an unyielding coal body into a yielding state, causing rapid softening of coal bodies in the extreme regions of postpeak stress and direct destruction of undamaged coal bodies, while the weak-energy disturbance does not have this characteristic. It can be seen that the instability of the coal body induced by a strong disturbance is not affected by the critical state and that the concept of the critical state only has practical significance for weak-energy disturbances. (4) The process of instability caused by dynamic disturbance is different. The coal body under strong disturbance conditions undergoes high-energy instantaneous destruction, and its action time is extremely short, which presents strict requirements for process monitoring and information acquisition. The effect of weak-energy disturbances on the instability of the coal body is related to fatigue damage. Only when the damage develops to a certain extent does the destabilization phenomenon occur, and there is a relatively long accumulation process. This provides relatively favourable conditions for the prediction and early warning of rockbursts induced by weak-energy disturbances and disaster prevention.

## 5. Conclusions

In this study, the self-developed experimental platform for coupling static load and vibration load is used to elucidate the mechanism of "critical static stress + slight disturbance" leading to rockburst disasters. During the test, the differences in MS signals among different samples, loading modes and damage processes are systematically analysed, high-

lighting the possibility of using MS signals to reflect the overall course and key nodes of coal and rock destabilization induced by weak-energy and low-frequency disturbances. Using microseismic signals to record the entire process and key nodes of coal body failure will help to predict and give an early warning of rockburst accidents through acoustic and electrical means, ensuring safe production in coal mines. The following conclusions can be drawn:

(1) The three-dimensional longitudinal wave velocities of the briquette samples are relatively close to each other, indicating that the prepared coal samples have good homogeneity. The value of longitudinal wave velocity is affected by the combined effect of particle size, density and moulding pressure. Generally, the smaller the particle size is, the higher the density and forming pressure, the denser the sample and the higher the longitudinal wave velocity.

(2) The natural frequency of the briquette sample can be obtained by using the soft support and knocking method, and the monitored signals can be denoised by the HHT decomposition method. The excitation signal is a broad-frequency signal covering the range of 0~500 Hz without obvious peaks, and the response signal reflects that the natural frequencies of the briquette samples are located between 30.79~43.34 Hz and most of them fluctuate up and down at 35 Hz.

(3) In the static loading test, the continuous cluster appearance of more than three large amplitude MS signals is an important criterion for determining the critical failure of the sample. At this time, the energy of the MS signals and the load-bearing capacity of the briquette simultaneously reach a peak, indicating that the sample is in a critical state of damage and will immediately lose its load-bearing capacity if further loading is continued.

(4) Digital filtering superimposed with EMD denoising can effectively extract the MS signals under static and dynamic combination conditions, and the effective signals can reflect the overall process and key nodes of coal and rock failure and instability. The weak-energy and low-frequency disturbance actually leads to fatigue failure, and the briquette sample undergoes three stages: the near-threshold stage, high-speed expansion stage and final fracture stage. The smaller the particle size of the briquette sample is, the denser the specimen, the stronger the amplitude and energy of the single effective MS signal formed during the damage process of the coal sample, the longer the time for crack extension from the near-threshold stage to the high-speed expansion stage, and the stronger the ability of the coal sample to resist the weak-energy and low-frequency disturbance.

(5) There are significant differences between the weak-energy and low-frequency disturbances and strong disturbances in the degree of stress impact, structure damage mode, coal instability premise and induced instability process. Compared with strong disturbances, weak-energy and low-frequency disturbances are mostly close-contact operations and have a certain degree of concealment, which poses a greater threat to operators.

**Author Contributions:** Conceptualization, Y.H.; methodology, T.J.; software, J.X.; validation, C.L.; review and editing, J.P.; writing—original draft, X.S. All authors have read and agreed to the published version of the manuscript.

**Funding:** This research is supported by the Fundamental Research Program of Shanxi Province (202103021224277) and by the Scientific and Technological Innovation Programs of Higher Education Institutions in Shanxi (2021L334).

**Institutional Review Board Statement:** Not applicable.

**Informed Consent Statement:** Not applicable.

**Data Availability Statement:** The data used to support the findings of this study are available from the corresponding author upon request.

**Conflicts of Interest:** The authors declare that they have no conflict of interest regarding the publication of this study.

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
