# Peer review of "Microseismic Signal Characteristics of the Coal Failure Process under Weak-Energy and Low-Frequency Disturbance"

_sustainability, doi:10.3390/su151914387_

Round 1
Reviewer 1 Report
This article discusses current and important issues related to the study of the characteristics of the microseismic signal of the destruction process in mines.
Studying the causes of firedamp explosions is one of the priority areas of research in the field of ensuring fire and industrial safety in coal mines. Many of these reasons have already been studied and served as the basis for the development of recommendations for accident prevention and safety rules for mining operations. At the same time, due to the increasingly complex mining technology, new, previously insufficiently studied effects are increasingly manifested, leading to accidents and disasters, seemingly for no apparent reason.
Currently, facts of the occurrence of electromagnetic precursors of earthquakes are known. This is evidenced by the analysis of long-term continuous observations, which made it possible to detect a clear manifestation of electromagnetic radiation, acoustic emission and electromagnetic emission, which is consistent with the developing ideas about the processes of creep of rocks in the final phase of preparation of a tectonic earthquake. The information obtained about the occurrence of electromagnetic radiation is not related to breakdown voltages in the atmosphere, but does not contradict the model of a radiating skin layer during crack formation in rocks. A roof collapse can be classified as a mini-rockburst, and a rockburst as a local tectonic destruction of rocks accompanied by electromagnetic radiation.
At the same time, the nature of the occurrence of electromagnetic radiation still remains unexplored and requires further research and clarification. It can be assumed that breakdown voltages of the electric field arise during the destruction of rocks between the edges of cracks and initiate spark discharges, which in turn can serve as sources of fires and explosions of methane-air and dust-air mixtures in the mine atmosphere, including in zones of controlled rock destruction.
The results obtained in the article are of undoubted interest to readers in the field under consideration, and can be used in solving similar problems.
However, there are the following issues that should be clarified:
1. The introduction should expand the literature review on possible emergency situations that occur in coal mines and ways to prevent them, as well as rock bursts and their consequences. In particular, it was possible to give examples of the influence of methane, aerodynamic processes, etc., as can be seen from the following works:
https://doi.org/10.3390/fire6030095
https://doi.org/10.3390/fire5060186
2. The article could conduct a comparative analysis of damage occurring in different regions of the world, giving the percentage of the most frequent emergency situations for greater relevance and significance of the research presented in the work.
3. Are there plans to obtain a patent for the complex research scheme presented in Figures 2, 3?
4. A more detailed analysis of the experimental process design shown in Figure 4 should be provided.
5. According to the data presented in tables 1, 2, a regression analysis should be carried out with the presentation in the article of the corresponding mathematical models for calculating and predicting the output values under consideration.
6. In section “3.3.1. Evolution law of the MS signal amplitude and spectrum”, we should dwell in more detail on the mathematical processing of data using the software used.
7. What practical application do the values of the coefficients presented in Figure 9 have?
8. The conclusions should dwell in more detail on the prospects for further research and testing of the results obtained at similar production facilities.
9. The list of references should include more sources from scientists from around the world, in particular from European countries, as well as North and Latin America.
Author Response
Dear Reviewer:
We gratefully thank you for taking the time to express constructive remarks and useful suggestions, which has significantly raised the quality of the manuscript and has enable us to improve the manuscript. Each suggested revision and comment, you put forward has been accurately incorporated and considered. Below your comments are response point by point and the revisions are indicated.
General comments:
This article discusses current and important issues related to the study of the characteristics of the microseismic signal of the destruction process in mines.
Studying the causes of firedamp explosions is one of the priority areas of research in the field of ensuring fire and industrial safety in coal mines. Many of these reasons have already been studied and served as the basis for the development of recommendations for accident prevention and safety rules for mining operations. At the same time, due to the increasingly complex mining technology, new, previously insufficiently studied effects are increasingly manifested, leading to accidents and disasters, seemingly for no apparent reason.
Currently, facts of the occurrence of electromagnetic precursors of earthquakes are known. This is evidenced by the analysis of long-term continuous observations, which made it possible to detect a clear manifestation of electromagnetic radiation, acoustic emission and electromagnetic emission, which is consistent with the developing ideas about the processes of creep of rocks in the final phase of preparation of a tectonic earthquake. The information obtained about the occurrence of electromagnetic radiation is not related to breakdown voltages in the atmosphere, but does not contradict the model of a radiating skin layer during crack formation in rocks. A roof collapse can be classified as a mini-rockburst, and a rockburst as a local tectonic destruction of rocks accompanied by electromagnetic radiation.
At the same time, the nature of the occurrence of electromagnetic radiation still remains unexplored and requires further research and clarification. It can be assumed that breakdown voltages of the electric field arise during the destruction of rocks between the edges of cracks and initiate spark discharges, which in turn can serve as sources of fires and explosions of methane-air and dust-air mixtures in the mine atmosphere, including in zones of controlled rock destruction.
The results obtained in the article are of undoubted interest to readers in the field under consideration, and can be used in solving similar problems.
Reply:
On behalf of my co-authors, we are very grateful to you for giving us the opportunity to revise our manuscript. We have carefully studied your comments and revised the paper in the draft review mode. Relevant questions in the original manuscript, such as references, experimental process design and mathematical models, have been fully supplemented and explained in the revised version.
Thank you again for your valuable comments, which are instructive for our current manuscript and future research.
Revision list according to your comments:
Comment (1)
The introduction should expand the literature review on possible emergency situations that occur in coal mines and ways to prevent them, as well as rock bursts and their consequences. In particular, it was possible to give examples of the influence of methane, aerodynamic processes, etc., as can be seen from the following works:
https://doi.org/10.3390/fire6030095 https://doi.org/10.3390/fire5060186
Reply:
Thank you for your introduction to these wonderful research work. Your suggestions provide us with some keen scientific insight. We have already added these related references in the manuscript as References [18] and [19].
Comment (2)
The article could conduct a comparative analysis of damage occurring in different regions of the world, giving the percentage of the most frequent emergency situations for greater relevance and significance of the research presented in the work.
Reply:
We thank the Reviewer for the thoughtful comment, with which we fully agree. In the revised manuscript, we investigated the occurrence of rockbursts in various coal-mining countries in the world (e.g., the United States, the Czech Republic, Poland, South Africa, Canada, Australia, Germany, Russia, and China), which were attached as references to the manuscript, e.g., references [1]-[11]. Unfortunately, due to various reasons, the statistical caliber and time of the data in the above documents are not completely consistent, and some are only analyzed as a case study, so it is it is difficult to give the specific percentage for the time being. Our study presents a large number of rockburst accidents in China, and the research results are of great significance to other regions in the world.
Once again, we appreciate your valuable suggestions, and we will collect and collate relevant information in this field in future research.
Comment (3)
Are there plans to obtain a patent for the complex research scheme presented in Figures 2, 3?
Reply:
We totally understand the reviewer' s concern. The dynamic‒static combination test system has obtained a Chinese patent, the natural frequency testing system has not yet applied for a patent.
Comment (4)
A more detailed analysis of the experimental process design shown in Figure 4 should be provided.
Reply:
We gratefully appreciate for your comment. In the revised version, we have added necessary supplementary information to make the introduction of the testing process more comprehensive and systematic (Page 8, Lines 281-297).
Comment (5)
According to the data presented in tables 1, 2, a regression analysis should be carried out with the presentation in the article of the corresponding mathematical models for calculating and predicting the output values under consideration.
Reply:
Thanks for your valuable counsel. In the revised manuscript, we added graphical descriptions of Tables 1 and 2, as shown in Figure 5 and Figure 10. Figure 10 is a scatter plot of the natural frequency distribution of each sample. Figure 5 shows the relationship between the longitudinal wave velocity, particle size, density, and forming pressure. In Figure 5, we conducted a linear regression of the above parameters and explained the mathematical properties exhibited in full detail (Page 10, Lines 328-336).
Comment (6)
In section “3.3.1. Evolution law of the MS signal amplitude and spectrum”, we should dwell in more detail on the mathematical processing of data using the software used.
Reply:
Thank you for your constructive comment. The software used in the paper is independently developed by us based on the principles of EMD and the HHT algorithm. In the revised manuscript, we have added the EMD principle and computer flowchart, please check it on Figure 7 and Page 12, Lines 364-378 of the revised manuscript.
Comment (7)
What practical application do the values of the coefficients presented in Figure 9 have?
Reply:
We thank you for reminding us this important point. Figure 9 in the original article reflects the relationship between the energy of the microseismic signals and the stresses monitored by the jack. There are two reasons for our approach. Firstly, the microseismic signals in Figure 8 are actually directional vectors, and a direct comparison seems inappropriate. We convert them into energy with scalar properties, which will be more conducive to visually representing the changes in signal energy as the loading process progresses. Secondly, the stress monitored by the jack can reflect the bearing capacity of the coal body in real time. When the stress value instantly drops significantly, it is considered that the specimen has lost its stability at this time. Coal-rock instability is an important prerequisite for the occurrence of rockburst. Using microseismic signals to record the entire process and key nodes of coal body failure, will help to predict and early warning of rockburst accidents through acoustic and electrical means, ensuring safe production in coal mines.
Comment (8)
The conclusions should dwell in more detail on the prospects for further research and testing of the results obtained at similar production facilities.
Reply:
Thank you for your valuable feedback. In the conclusion section of the revised manuscript, we have added in-depth perspectives on the application prospects and future prospects of the research results (Page 31, Lines 768-770).
Comment (9)
The list of references should include more sources from scientists from around the world, in particular from European countries, as well as North and Latin America.
Reply:
Thank you for the excellent and insightful series of remarks. Compared with the original manuscript, we have added 21 references in the revised version, most of which are representative papers from scientists in Europe and America, which have benefited us greatly.
We appreciate for your warm work earnestly, and hope that the correction will meet with approval.
Once again, thank you very much for your comments and suggestions.

Reviewer 2 Report
This study provides technical support and theoretical guidance for using microseismic methods to monitor the induced effects of weak-energy and low-frequency disturbances on rockburst. This study is interesting. The results of this study are of positive significance, and the conclusions are quite convincing. The following comments are provided for further improvement during the revision.
1. Section 2.1:
The artificial briquette used in this paper is an excellent method to avoid other factors and focus on the effects of sample size, density, and molding pressure. However, for the preparation process and precautions of the briquette, the description is too simple, and the preparation of the above samples is not introduced in detail, which will confuse the readers.
2. Section 2.2:
The text in Figure 3 is not very clear. It is suggested to enlarge it appropriately or make adjustments using other methods.
3. Section 3:
The focus of this study is to consider the damage and failure model of a coal-rock mass under weak energy and low-frequency disturbances. However, during the model validation process, only coal was utilized without considering rock. Coal and rock exhibit significant differences in mechanical properties. Therefore, it is recommended to modify the term "coal-rock mass" in the experimental section of the paper or include additional validation involving rock-related considerations.
4. Section 4:
The author did a good experiment, and some of the phenomena presented are also very interesting. However, the discussion section of the manuscript should be simple and focus on:
(1) Whether the difference between weak-energy disturbances and strong disturbances has been comprehensively described;
(2) If possible, compare the causes of these phenomena with the relevant research results.
5. References
The following references are suggested to be cited: " Weak disturbance-triggered seismic events: an experimental and numerical investigation. Bulletin of Engineering Geology and the Environment, 2019, 78: 2943-2955. "; "Experimental study on the failure characteristics of granite subjected to weak dynamic disturbance under different σ3 conditions. Rock Mechanics and Rock Engineering, 2021, 54: 5577-5590."
Minor editing of English language required
Author Response
Response to Reviewer 2:
Thank you very much for your kindly comments on our manuscript. There is no doubt that these comments are valuable and very helpful for revising and improving our manuscript. In what follows, we would like to answer the questions you mentioned and give detailed account of the changes made to the original manuscript.
General comments:
This study provides technical support and theoretical guidance for using microseismic methods to monitor the induced effects of weak-energy and low-frequency disturbances on rockburst. This study is interesting. The results of this study are of positive significance, and the conclusions are quite convincing. The following comments are provided for further improvement during the revision.
Reply:
Your comments are all valuable and very helpful for revising and improving our paper. Thank you very much for your subsequent specific suggestions. We have studied comments carefully and have made correction which we hope meet with approval. Please check it again.
Revision list according to your comments:
Comment (1) Section 2.1
The artificial briquette used in this paper is an excellent method to avoid other factors and focus on the effects of sample size, density, and molding pressure. However, for the preparation process and precautions of the briquette, the description is too simple, and the preparation of the above samples is not introduced in detail, which will confuse the readers.
Reply:
Thank you for your significant reminding. According to your suggestion, we have provided detailed instructions on the preparation process of the briquette samples. Please check it on Page 4, Lines 180-189 of the revised manuscript.
Comment (2) Section 2.2:
The text in Figure 3 is not very clear. It is suggested to enlarge it appropriately or make adjustments using other methods.
Reply:
We thank you for reminding us this important point. In the revised version, we have re-provided high-definition graphics to ensure that the resolution ratio of color maps is not less than 300 DPI, and the resolution ratio of dot and line maps is not less than 600 DPI. We hope our revised manuscript can meet the journal’s standard.
Comment (3) Section 3:
The focus of this study is to consider the damage and failure model of a coal-rock mass under weak energy and low-frequency disturbances. However, during the model validation process, only coal was utilized without considering rock. Coal and rock exhibit significant differences in mechanical properties. Therefore, it is recommended to modify the term "coal-rock mass" in the experimental section of the paper or include additional validation involving rock-related considerations.
Reply:
Thank you for your careful scrutiny. We have corrected the error. Please check it in the revised manuscript.
Comment (4) Section 4:
The author did a good experiment, and some of the phenomena presented are also very interesting. However, the discussion section of the manuscript should be simple and focus on:
(1) Whether the difference between weak-energy disturbances and strong disturbances has been comprehensively described;
(2) If possible, compare the causes of these phenomena with the relevant research results.
Reply:
We greatly appreciate your insightful comments. In the revised manuscript, we have read through the discussion section and are confident that we have fully described the distinction between weak-energy disturbances and strong disturbances. In addition, the causes of the above phenomena have been fully compared with relevant research results.
Comment (5) References:
The following references are suggested to be cited: " Weak disturbance-triggered seismic events: an experimental and numerical investigation. Bulletin of Engineering Geology and the Environment, 2019, 78: 2943-2955. "; "Experimental study on the failure characteristics of granite subjected to weak dynamic disturbance under different σ3 conditions. Rock Mechanics and Rock Engineering, 2021, 54: 5577-5590."
Reply:
We thank you for reminding us this important point. We have added the related documents in the manuscript as References [41] and [48].
In addition, we are very sorry for the mistakes in this manuscript and the inconvenience they caused during your reading. We have revised the whole manuscript carefully and tried to avoid the grammar or syntax error. In addition, the manuscript has been edited by American Journal Experts (AJE) workshop for language service, and a copy of the editing certificate is provided as part of the cover letter. We believe that the language is now acceptable for the review process.
We hope that the changes we have made resolve all your concerns about the article. We are more than happy to make any further changes that will improve the paper and/or facilitate successful publication.

Round 2
Reviewer 1 Report
The authors have revised the article. Thanks to the authors for their work. The article may be published.